# Identification of a basal system for unwinding a bacterial chromosome origin

Tomas T Richardson[1], Daniel Stevens[1,†], Simone Pelliciari[1,†] (ID), Omar Harran[1], Theodor Sperlea[2] (ID) & Heath Murray[1,*] (ID)

## Abstract

Genome duplication is essential for cell proliferation, and DNA synthesis is generally initiated by dedicated replication proteins at specific loci termed origins. In bacteria, the master initiator DnaA binds the chromosome origin (*oriC*) and unwinds the DNA duplex to permit helicase loading. However, despite decades of research it remained unclear how the information encoded within *oriC* guides DnaA-dependent strand separation. To address this fundamental question, we took a systematic genetic approach *in vivo* and identified the core set of essential sequence elements within the *Bacillus subtilis* chromosome origin unwinding region. Using this information, we then show *in vitro* that the minimal replication origin sequence elements are necessary and sufficient to promote the mechanical functions of DNA duplex unwinding by DnaA. Because the basal DNA unwinding system characterized here appears to be conserved throughout the bacterial domain, this discovery provides a framework for understanding *oriC* architecture, activity, regulation and diversity.

**Keywords** DNA replication; DnaA; origin; unwinding
**Subject Categories** DNA Replication, Repair & Recombination
**The EMBO Journal (2019) 38: e101649**

## Introduction

Accurate transmission of genetic material is a fundamental requirement for the viability of all cells. In most cases, DNA replication must commence once (and only once) per cell cycle to ensure rigorous coordination of genome duplication and segregation. Dysfunction of DNA replication initiation can lead to improper chromosome inheritance, disease and cell death.

Throughout the domains of life, conserved proteins containing AAA+ (ATPase Associated with various cellular Activities) domains assemble into dynamic multimeric complexes on double-stranded DNA (dsDNA) and direct loading of the replicative helicase (Bleichert *et al*, 2017). Subsequent helicase activation promotes assembly of the replication machinery that catalyses DNA synthesis. In bacteria, the ring-shaped hexameric helicases that drive bidirectional replication from a chromosome origin (*oriC*) are loaded around single-stranded DNA (ssDNA). Bacterial chromosome origin unwinding to permit helicase loading proceeds through a mechanism involving the ubiquitous master initiation protein DnaA.

DnaA is a multifunctional enzyme composed of four distinct domains that act in concert during DNA replication initiation (Appendix Fig S1A; Messer *et al*, 1999). Domain IV contains a helix-turn-helix dsDNA binding motif that specifically recognizes 9 base pair asymmetric sequences called "DnaA-boxes" (consensus 5′-TTATCCACA-3′; Appendix Fig S1B; Fuller *et al*, 1984; Roth & Messer, 1995; Fujikawa *et al*, 2003). Domain III is composed of the AAA+ motif that can assemble into an ATP-dependent right-handed helical filament (Appendix Fig S1C) (Schaper & Messer, 1997; Erzberger *et al*, 2002, 2006). Domain III also contains the residues required for DnaA to interact specifically with a ssDNA binding site termed the "DnaA-trio" (Ozaki *et al*, 2008; Duderstadt *et al*, 2011; Richardson *et al*, 2016). The DnaA-trio is a repeating trinucleotide motif (consensus 3′-GAT-5′) originally discovered as an essential element within the *Bacillus subtilis* origin and then identified in origins throughout the bacterial domain. It has been proposed that DnaA-trios stabilize a DnaA filament on a single DNA strand, with each DnaA-trio motif interacting with a single subunit of DnaA from the filament (Richardson *et al*, 2016). Domain II tethers domains III-IV to domain I, and domain I acts as an interaction hub that facilitates DnaA oligomerization (Weigel *et al*, 1999) and loading of the replicative helicase (Sutton *et al*, 1998).

Bacterial replication origins encode information that promotes specific unwinding of the DNA duplex by DnaA (Bramhill & Kornberg, 1988; Wolanski *et al*, 2014). Typically bacterial origins contain multiple DnaA-boxes, flanked by the DNA unwinding site which often contains DnaA-trios and an intrinsically unstable AT-rich element (Fuller *et al*, 1984; Kowalski & Eddy, 1989; Richardson *et al*, 2016). However, comparison of *oriC* regions from throughout the bacterial domain shows that they are highly diverse, containing a variable number and distribution of DnaA-boxes (Mackiewicz *et al*, 2004; Luo *et al*, 2018). This heterogeneity likely reflects the

1 Centre for Bacterial Cell Biology, Institute for Cell and Molecular Biosciences, Newcastle University, Newcastle Upon Tyne, UK
2 Chromosome Biology Group, LOEWE Center for Synthetic Microbiology, SYNMIKRO, Philipps-Universität Marburg, Marburg, Germany
*Corresponding author. Tel: +44 1912083233; E-mail: heath.murray@newcastle.ac.uk
†These authors contributed equally to this work

range of regulatory networks employed by bacteria to coordinate DNA synthesis during their cell cycle in response to different growth conditions and environments. Significantly, this complexity has obscured identification of the core set of sequences that constitute a basal bacterial origin unwinding system.

The binding of DnaA to DnaA-boxes is thought to organize formation of a nucleoprotein complex that mechanically separates the DNA duplex. Several models for DnaA activity at *oriC* have been proposed: (i) altering DNA topology, (ii) DNA stretching and (iii) ssDNA recruitment. First, the presence of intrinsically unstable DNA unwinding elements within chromosome origins (Kowalski & Eddy, 1989) suggests that DnaA may act by altering dsDNA topology to promote spontaneous opening of these AT-rich sequences. Consistent with this model, electron microscopy and DNase footprinting studies suggest that DnaA wraps dsDNA around the outside of its nucleoprotein complex at *oriC* (Fuller *et al*, 1984; Funnell *et al*, 1987), while structural, biochemical and biophysical studies indicate that a right-handed DNA wrap is formed around the DnaA$^{ATP}$ filament (Erzberger *et al*, 2006; Zorman *et al*, 2012). Second, the ability of the DnaA$^{ATP}$ to specifically bind and stretch ssDNA (Speck & Messer, 2001; Duderstadt *et al*, 2011; Cheng *et al*, 2015; Richardson *et al*, 2016) suggests that a DnaA filament could promote origin unwinding by assembling onto one strand of dsDNA and destabilizing the double helix. Third, a combination of biochemical, genetic and modelling data using the *Escherichia coli* replication system suggests that DnaA binds dsDNA as a monomer proximal to the unwinding region and as an oligomer distal to the unwinding region (Ozaki & Katayama, 2012a; Kaur *et al*, 2014; Noguchi *et al*, 2015; Shimizu *et al*, 2016; Sakiyama *et al*, 2017). Subsequent binding of an architectural protein between the proximal and distal sites introduces a sharp bend in the dsDNA that connects DnaA proteins from the two regions to promote unwinding and recruit ssDNA to the dsDNA-bound DnaA oligomer. Note that aspects of these various mechanisms could act in conjunction with each other depending upon the organization of a particular *oriC*.

A key outstanding issue in the field is whether any of the proposed models for bacterial chromosome origin unwinding are physiologically relevant. Structure/function analysis of *oriC in vivo* is challenging because the locus is required for viability; mutation of an essential feature will be lethal, while mutations that severely disable *oriC* can rapidly accumulate compensatory suppressors. For this reason, cloned *oriC* fragments that support plasmid replication were historically used to analyse origin sequences (Oka *et al*, 1980). Such studies using the well-characterized *E. coli oriC* plasmids indicated that a significant number of the DnaA-boxes and binding sites for architectural proteins were essential for origin activity (Asai *et al*, 1990; Woelker & Messer, 1993; Roth *et al*, 1994; Langer *et al*, 1996; McGarry *et al*, 2004). However, several laboratories subsequently showed that this plasmid system does not faithfully reflect *oriC* within its native environment and mutagenesis of the endogenous *E. coli oriC* has revealed that none of the protein binding sites is individually essential for origin function (Weigel *et al*, 2001; Stepankiw *et al*, 2009; Kaur *et al*, 2014; Noguchi *et al*, 2015; Sakiyama *et al*, 2017). Thus, despite isolation of a bacterial replication origin almost forty years ago, the minimal essential sequences that are necessary and sufficient to support a specific DnaA-dependent unwinding mechanism remain unknown.

In this study, we sought to address this long-standing and fundamental question. To help overcome challenges related to the genetic complexity of bacterial replication origins, we exploited the observation that some bacterial species contain a naturally bipartite *oriC*. It has been shown in both *Bacillus subtilis* and *Helicobacter pylori* that while both halves of their bipartite origins are essential *in vivo*, only one of these regions is required for DNA unwinding *in vitro* (Moriya *et al*, 1992; Krause *et al*, 1997; Donczew *et al*, 2012). We hypothesized that this partitioning of functions would simplify the search for sequences within the chromosome origin unwinding region that are needed for both *oriC* function *in vivo* and DnaA-dependent DNA strand separation *in vitro*.

# Results

## A subset of DnaA-boxes within *incC* are essential for origin activity

The unwinding region of the bipartite *B. subtilis oriC* is located between the *dnaA* and *dnaN* genes and has been termed *incC* (from the observation that plasmids containing this region inhibited chromosomal replication initiation, thus displaying "partial incompatibility"; note that throughout the present study, we will refer to the activity of *incC* as supporting DNA unwinding within the endogenous origin; Appendix Fig S2; Moriya *et al*, 1988). The *incC* region contains seven DnaA-boxes and the essential DnaA-trio motifs (Fig 1A; Fukuoka *et al*, 1990; Richardson *et al*, 2016).

To enable identification of essential DnaA-boxes within *incC* without selecting for suppressor mutations, we utilized a strain in which DNA replication can conditionally initiate from a plasmid origin (*oriN*) integrated into the chromosome (Fig 1B) (Richardson *et al*, 2016). Activity of *oriN* requires its cognate initiator protein, RepN; both of these factors act independently of *oriC*/DnaA (Hassan *et al*, 1997). Expression of *repN* was placed under the control of an isopropyl-β-D-thiogalactoside (IPTG)-inducible promoter, thus permitting both the introduction of mutations into *incC* and their subsequent analysis following removal of the inducer to deplete *oriN* activity.

Previous work identified the consensus DnaA-box#6 as the only DnaA-box necessary for origin function (Richardson *et al*, 2016). Therefore, we wondered whether DnaA-box#6 might also be sufficient for *incC* activity. Using an *incC* targeting vector (Appendix Fig S2), the sequences of DnaA-box#1/2/3/4/5/7 were scrambled using site-directed mutagenesis and the resulting plasmid was integrated into the chromosome by double-homologous recombination. Figure 1C shows that this origin is not functional, indicating that DnaA-box#6 is not sufficient for *incC* activity.

To identify the additional DnaA-boxes required for activity of the origin unwinding region, binding sites were sequentially mutated starting either from the distal or proximal end of *incC* (relative to the DnaA-trios) or from the middle. Note that this approach maintained the correct spacing between all of the remaining sequences. From the proximal end, mutation of DnaA-box#6/7 was lethal (Fig 1D), while from the distal end, disruption of DnaA-box#1/2/3 abated origin function (Fig 1E). In contrast, a strain lacking DnaA-box#4/5 from the middle was viable (Fig 1F). Therefore, we used the ΔDnaA-box#4/5 vector as a starting point to construct additional

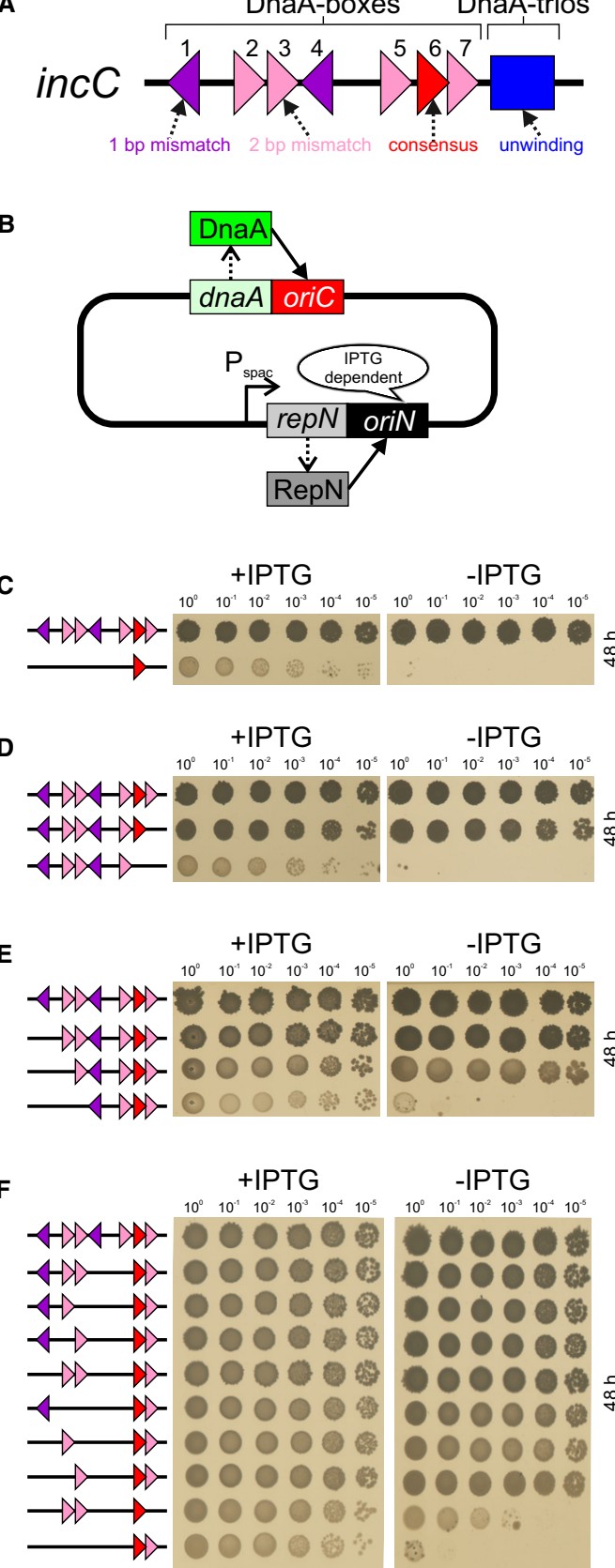

**Figure 1.**

**Figure 1. Subsets of essential DnaA-boxes within the replication origin unwinding region.**

A   The *B. subtilis incC* unwinding region. DnaA-boxes are numbered. Colouring indicates conservation relative to consensus (5′-TTATCCACA-3′ in red).

B   The *oriC*-independent strain used for constructing chromosome origin and *dnaA* mutants.

C   DnaA-box#6 is not sufficient for *incC* activity. Wild-type (HM1552), *incC*ΔDnaA-box#1/2/3/4/5/7 (HM1607).

D   Mutagenesis of DnaA-boxes proximal to the DnaA-trios indicates that DnaA-box#6/7 are necessary for *incC* activity. Wild-type (HM1552), *incC*ΔDnaA-box#7 (HM1651), *incC*ΔDnaA-box#6/7 (HM1554).

E   Mutagenesis of DnaA-boxes distal to the DnaA-trios indicates that DnaA-box#1/2/3 are necessary for *incC* activity. Wild-type (HM1108), *incC*ΔDnaA-box#1 (OH21), *incC*ΔDnaA-box#1/2 (OH61), *incC*ΔDnaA-box#1/2/3 (OH62).

F   Sequential mutagenesis of DnaA-boxes within *incC* indicates that two sets of DnaA-boxes, one proximal and one distal to the DnaA-trios, are required for chromosome origin function. Wild-type (HM1108), *incC*ΔDnaA-box#4/5 (TR452), *incC*ΔDnaA-box#3/4/5 (TR411), *incC*ΔDnaA-box#2/4/5 (TR412), *incC*ΔDnaA-box#1/4/5 (TR124), *incC*ΔDnaA-box#2/3/4/5 (HM1671), *incC*ΔDnaA-box#1/3/4/5 (TR122), *incC*ΔDnaA-box#1/2/4/5 (TR123), *incC*ΔDnaA-box#1/4/5/7 (HM1643), *incC*ΔDnaA-box#1/2/3/4/5 (OH64).

mutants with the aim of identifying a minimal set of DnaA-boxes capable of supporting *incC* activity. Mutagenesis of the three upstream DnaA-boxes (#1/2/3) showed that each could be individually inactivated and that all double-mutant combinations were tolerated (Fig 1F). Critically however, constructs lacking a distal DnaA-box (ΔDnaA-box#1/2/3/4, ΔDnaA-box#1/2/3/4/5, ΔDnaA-box#1/2/3/4/7) were lethal (Fig 1F and Appendix Fig S3A). These results indicate that *incC* contains two functional DnaA-box subregions, one distal and one proximal to the DnaA-trios. Supporting this interpretation, we were also able to construct a minimal *incC* with partial activity containing a distinct two subregion structure with DnaA-box#2/3 upstream and DnaA-box#6 downstream (Fig 1F).

## Moving the upstream DnaA-box suggests it functions through a DNA loop

To begin investigating the essential role of the distal *incC* subregion, a series of variants was created with a rightward-facing consensus DnaA-box (DnaA-box#CR) introduced at different positions (Fig 2A). For these constructs, the entire region upstream of DnaA-box#6 was replaced by an artificial sequence that does not contain any other DnaA-boxes (note that this also removes any other specific DNA binding sites that may have been present). Beginning with a site located 22 base pairs upstream of DnaA-box#6, the DnaA-box#CR sequence was moved upstream by increments of five or six base pairs (i.e. ~half turns about the DNA double helix). Origin activity required that DnaA-box#CR was located at least 44 base pairs upstream of DnaA-box#6, which corresponds to the position of DnaA-box#3 within the native *incC* (Fig 2A, dotted vertical line). Interestingly, as the DnaA-box was moved further upstream its ability to support origin activity displayed phasing, with functional sites located on the same face of the DNA helix for 3 turns. Beyond this point, the effect of phasing was still detectable, although apparently no longer essential for origin activity.

Additional series of constructs were created to further characterize the properties of the upstream *incC* subregion. Using the same artificial backbone described above, a leftward-facing consensus DnaA-box (DnaA-box#CL) was introduced at several sites. Again, these upstream DnaA-boxes exhibited helical phasing (Fig 2B). Next, the upstream DnaA-box#CR was shifted significantly further upstream. In order to move the DnaA-box without interfering with *dnaA* gene expression, a larger artificial DNA sequence was introduced upstream of DnaA-box#6 (Fig 2C). As expected, in the absence of a distal DnaA-box this artificial sequence did not support origin function (Fig 2D). However, replication initiation activity

could be recovered when DnaA-box#CR was present 132, 297 or 462 base pairs upstream from DnaA-box#6 (Fig 2D).

Taken together, the following observations suggest that the distal DnaA-box might act through a DNA loop: (i) it can function at multiple locations; (ii) it can function in opposite orientations; (iii) it can function at great distances from the downstream subregion; and (iv) its activity displays helical phasing (Hochschild & Ptashne, 1986; Bellomy *et al*, 1988). The architecture of any DNA loop mediated by DnaA presumably would have to be relatively flexible to accommodate the many permissive locations of the upstream DnaA-box. In contrast, we found that *incC* was inactive if the orientation of either DnaA-box#6 or DnaA-box#6/7 was reversed (Appendix Fig S3B), suggesting that the location of the downstream DnaA-boxes is more restricted.

## Overexpression of DnaA rescues deletion of the upstream DnaA-box subregion

DNA looping has been proposed to play diverse roles in gene regulation and chromosome organization (Cournac & Plumbridge, 2013). The observed variability in positioning permitted for the distal DnaA-box suggested that the postulated DNA loop might be acting to produce a high local concentration of DnaA at the downstream subregion of *incC* (Mossing & Record, 1986; Oehler & Muller-Hill, 2010). A key prediction of this model is that removal of distal DnaA-boxes should be overcome by increasing the level of DnaA in the cell (Oehler *et al*, 1990). To test this, the *dnaA-dnaN* operon (lacking *incC*) was cloned downstream of a xylose-inducible promoter and introduced at an ectopic locus within the *oriN* strain (Fig EV1A). *dnaN* was included in this cassette because overexpression of DnaA has been shown to auto-repress expression of the endogenous *dnaA-dnaN* operon, leading to depletion of DnaN and defects in replication elongation (Ogura *et al*, 2001). Subsequently, an *incC* vector containing only DnaA-box#6/7 (*incC*$^{art}$ DnaA-box#6/7) was used to transform this strain. Immunoblots showed that in the presence of xylose, DnaA and DnaN levels were significantly increased (Fig EV1B). Critically, overexpression of DnaA and DnaN was able to rescue activity of the *incC* variant containing only DnaA-box#6/7 (Fig EV1C). A strain overexpressing only DnaA was also found to support activity of an *incC* containing DnaA-box#6/7 (Fig EV1D and E), confirming that DnaA alone was responsible for the observed rescue. These results are consistent with the model that distal DnaA-boxes within *incC* act to increase the local concentration of DnaA at the downstream subregion through DNA looping.

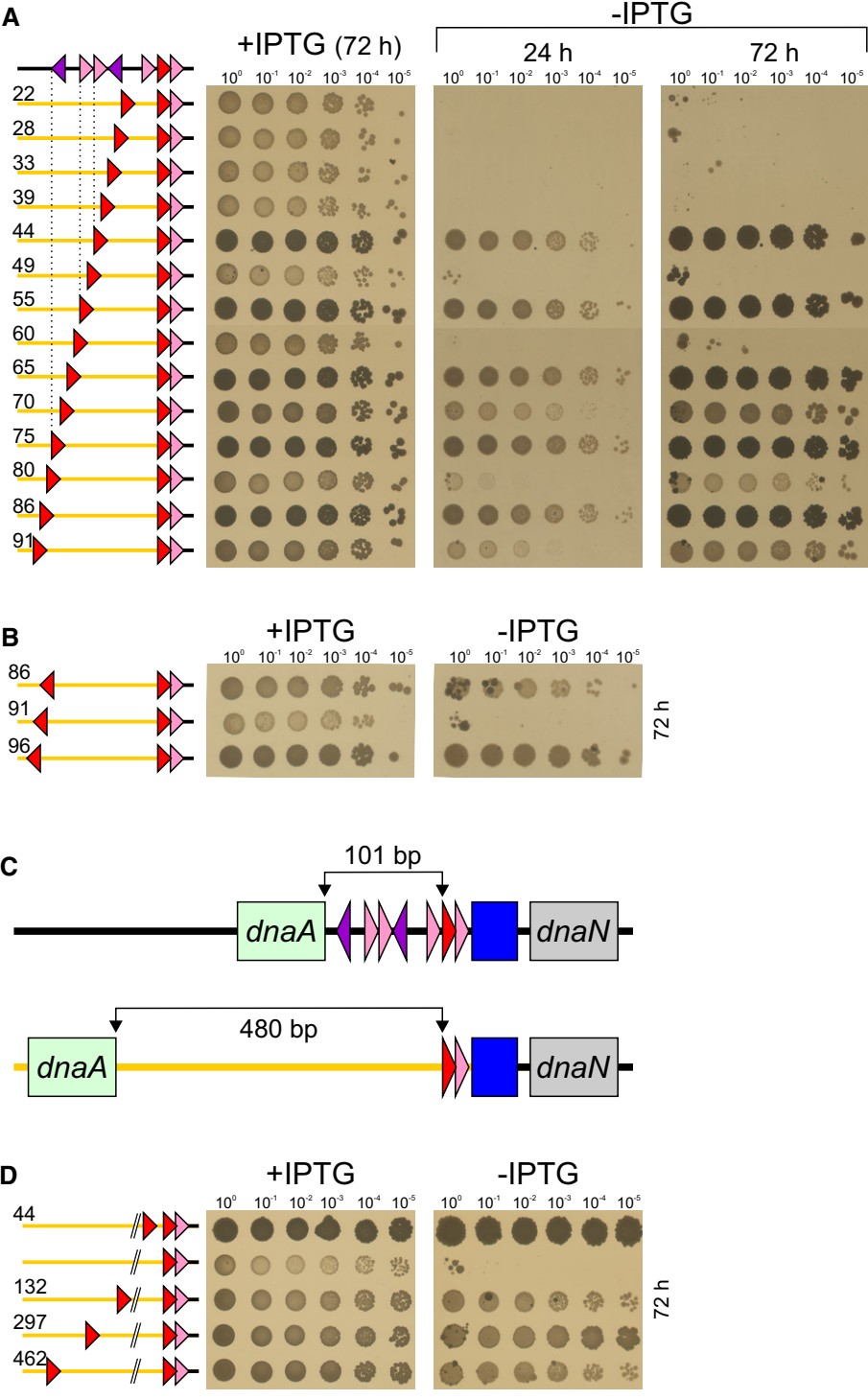

**Figure 2.**

To characterize this minimal *incC* variant with just DnaA-box#6/7, we constructed a strain lacking *oriN* to ensure that any residual activity from this origin would not complicate the analysis (Fig 3A). As expected, the viability of this strain required xylose to overexpress DnaA (Fig 3B and C). We found that the xylose dependency was retained even when the strain was growing slowly in minimal media, indicating that the requirement for a high local concentration

of DnaA is not solely for multifork replication during rapid growth (Appendix Fig S4).

Although overexpression of DnaA promotes sufficient activity from the minimal *incC* for viability, we observed that these strains display a growth defect (Fig 3B and C, and Appendix Fig S4). To investigate the likely cause of this phenotype, we visualized chromosomal DNA within live cells using the non-specific DNA-binding

◀

**Figure 2. A distal DnaA-box displays phasing and flexibility.**

A  Systematically shifting a distal rightward-facing consensus DnaA-box (CR) by half turns of the DNA double helix. For all constructs, the native 101 bp DNA upstream of DnaA-box#6 was replaced by a synthetic sequence ("$incC^{art}$" orange line). Numbering indicates the distance from the middle of DnaA-box#6 to the middle of the distal DnaA-box. $incC^{art}$DnaA-box#CR$^{22}$/6/7 (TR711), $incC^{art}$DnaA-box#CR$^{28}$/6/7 (TR710), $incC^{art}$DnaA-box#CR$^{33}$/6/7 (TR326), $incC^{art}$DnaA-box#CR$^{39}$/6/7 (TR325), $incC^{art}$DnaA-box#CR$^{44}$/6/7 (OH70), $incC^{art}$DnaA-box#CR$^{49}$/6/7 (TR324), $incC^{art}$DnaA-box#CR$^{55}$/6/7 (TR186), $incC^{art}$DnaA-box#CR$^{60}$/6/7 (TR328), $incC^{art}$DnaA-box#CR$^{65}$/6/7 (TR329), $incC^{art}$DnaA-box#CR$^{70}$/6/7 (TR330), $incC^{art}$DnaA-box#CR$^{75}$/6/7 (TR184), $incC^{art}$DnaA-box#CR$^{80}$/6/7 (TR709), $incC^{art}$DnaA-box#CR$^{86}$/6/7 (TR708), $incC^{art}$DnaA-box#CR$^{91}$/6/7 (TR707).

B  Shifting a leftward-facing consensus DnaA-box (CL) by half turns of the DNA double helix. $incC^{art}$DnaA-box#CL$^{86}$/6/7 (TR657), $incC^{art}$DnaA-box#CL$^{91}$/6/7 (TR690), $incC^{art}$DnaA-box#CL$^{96}$/6/7 (TR691).

C  Schematic diagram indicating the replacement of the native DNA sequence between *dnaA* and DnaA-box#6 with a larger synthetic sequence ("$incC^{x\_art}$").

D  The distal rightward-facing consensus DnaA-box (CR) can be shifted several hundred base pairs upstream of its endogenous location. $incC^{x\_art}$DnaA-box#CR$^{44}$/6/7 (TR210), $incC^{x\_art}$DnaA-box#6/7 (TR209), $incC^{x\_art}$DnaA-box#CR$^{132}$/6/7 (TR203), $incC^{x\_art}$DnaA-box#CR$^{297}$/6/7 (TR206), $incC^{x\_art}$DnaA-box#CR$^{462}$/6/7 (TR208).

protein HBsu-GFP as a reporter. In contrast to the strain with a wild-type origin where chromosomes were well segregated in each cell, the minimal *incC* strain appeared to have fewer chromosomes per cell and generated progeny lacking DNA (Fig 3D). Thus, the minimal *incC* does not appear capable of achieving a wild-type level of replication.

We wondered whether the disconnect between origin activity and cell growth observed in the minimal *incC* strain would preclude cellular differentiation. During nutrient deprivation, *B. subtilis* can undergo a developmental process of endospore formation to generate a dormant cell that is highly recalcitrant to environmental stresses (Piggot, 1996). Two chromosomes are required for a *B. subtilis* cell to complete endospore formation. If DNA replication is incomplete or the chromosomes are damaged, then sporulation is blocked (Lenhart *et al*, 2012). Strains were grown in sporulation medium, and the number of heat-resistant spores relative to total colony-forming units was determined. Initially, it appeared that sporulation was completely blocked in the minimal *incC* strain (Fig 3E). However, we noticed that there was also a ~100-fold decrease in sporulation of the parental strain with wild-type *incC* (Fig 3E). This suggested that the overexpression of DnaA might be inhibiting sporulation. It is known that DnaA activates the cell cycle checkpoint protein Sda to coordinate DNA synthesis with the onset of sporulation (Burkholder *et al*, 2001; Veening *et al*, 2009). Indeed, deletion of *sda* fully rescued the sporulation efficiency of the parental strain with wild-type *incC* and revealed that the strain with the minimal *incC* was also capable of sporulating, albeit at a lower efficiency (Fig 3E and F). This shows that the minimal *incC* ($incC^{art}$DnaA-box#6/7) can support a complex developmental process such as cellular differentiation.

**Activity of the DnaA protein delivered from the upstream subregion requires residues involved in filament formation and ssDNA binding**

The systematic genetic analysis of *incC* indicates that DnaA-box#6/7 are the only specific dsDNA recognition sequences required to support origin unwinding activity (provided that the DnaA expression level is high). These data, together with previous results (Duderstadt *et al*, 2011; Cheng *et al*, 2015; Richardson *et al*, 2016), are most compatible with a model for DNA duplex unwinding where DnaA-box#6/7 promotes DnaA filament formation onto the adjacent DnaA-trios to promote strand separation by ssDNA stretching.

If the upstream DnaA-box in *incC* functions to increase the local concentration of DnaA at the site of DNA unwinding and the strand

separation mechanism involves DnaA filaments stretching DnaA-trios, then this predicts that the DnaA protein being delivered from the upstream subregion would require both filament formation and ssDNA binding activities. To test this hypothesis, a DnaA chimera was created to analyse the functions of DnaA specifically required when bound at a distal DnaA-box. Most bacterial species, including *B. subtilis*, recognize the consensus DnaA-box sequence 5′-TTATC-CACA-3′. However, it has been found that the dsDNA binding domain of *Thermotoga maritima* DnaA contains several amino acid substitutions that alter its preferred DnaA-box recognition sequence (5′-AAACCTACCACC-3′; Ozaki *et al*, 2006). This variance presents the opportunity to create a DnaA chimera composed of *B. subtilis* domains I-III and *T. maritima* domain IV that binds specifically to the *T. maritima* DnaA-box but retains the native domains required for filament formation and ssDNA binding (Fig 4A; Noguchi *et al*, 2015).

To establish this system in *B. subtilis,* the *T. maritima* DnaA-box sequence (DnaA-box#Tm) was introduced into the artificial *incC* construct (containing DnaA-box#6/7) at the position corresponding to the native DnaA-box#3 and this vector was used to replace the endogenous *incC* region in the inducible *oriN* strain (Fig 4B and C). Next, a chimeric wild-type *dnaA* construct was cloned under the control of a xylose-dependent promoter and integrated at an ectopic locus (Fig 4B). In the absence of xylose, the hybrid origin was non-functional, showing that the native DnaA protein did not recognize the *T. maritima* DnaA-box (Fig 4C). Addition of xylose to induce the wild-type DnaA chimera rescued viability only when the artificial *incC* contained DnaA-box#Tm, demonstrating that the wild-type chimeric protein is capable of providing the required activity specifically at the distal binding site (Fig 4C).

Previous studies using *E. coli, T. maritima* and *Aquifex aeolicus* have implicated multiple DnaA residues in ssDNA binding (Fig EV2A; Ozaki *et al*, 2008; Duderstadt *et al*, 2011) and filament formation (Fig EV3A; Duderstadt *et al*, 2010; Ozaki *et al*, 2012b). To determine which of these amino acids are essential for *B. subtilis* DnaA activity, the endogenous *dnaA* gene was subjected to site-directed mutagenesis and the function of DnaA variants was characterized using the inducible *oriN* system. We found that two residues in the ssDNA binding motif (I190 and K222; Fig EV2B) and eight within the AAA+/AAA+ filament interface (R202, R206, F218, H231, R264, L269, G317 and R321; Fig EV3B) were essential for DnaA activity. Immunoblot analysis showed that defective DnaA proteins were stable and expressed at near wild-type levels (Figs EV2C and EV3C).

Lethal mutations were introduced into the *dnaA* chimera, and the alleles were transformed into the strain with DnaA-box#Tm located

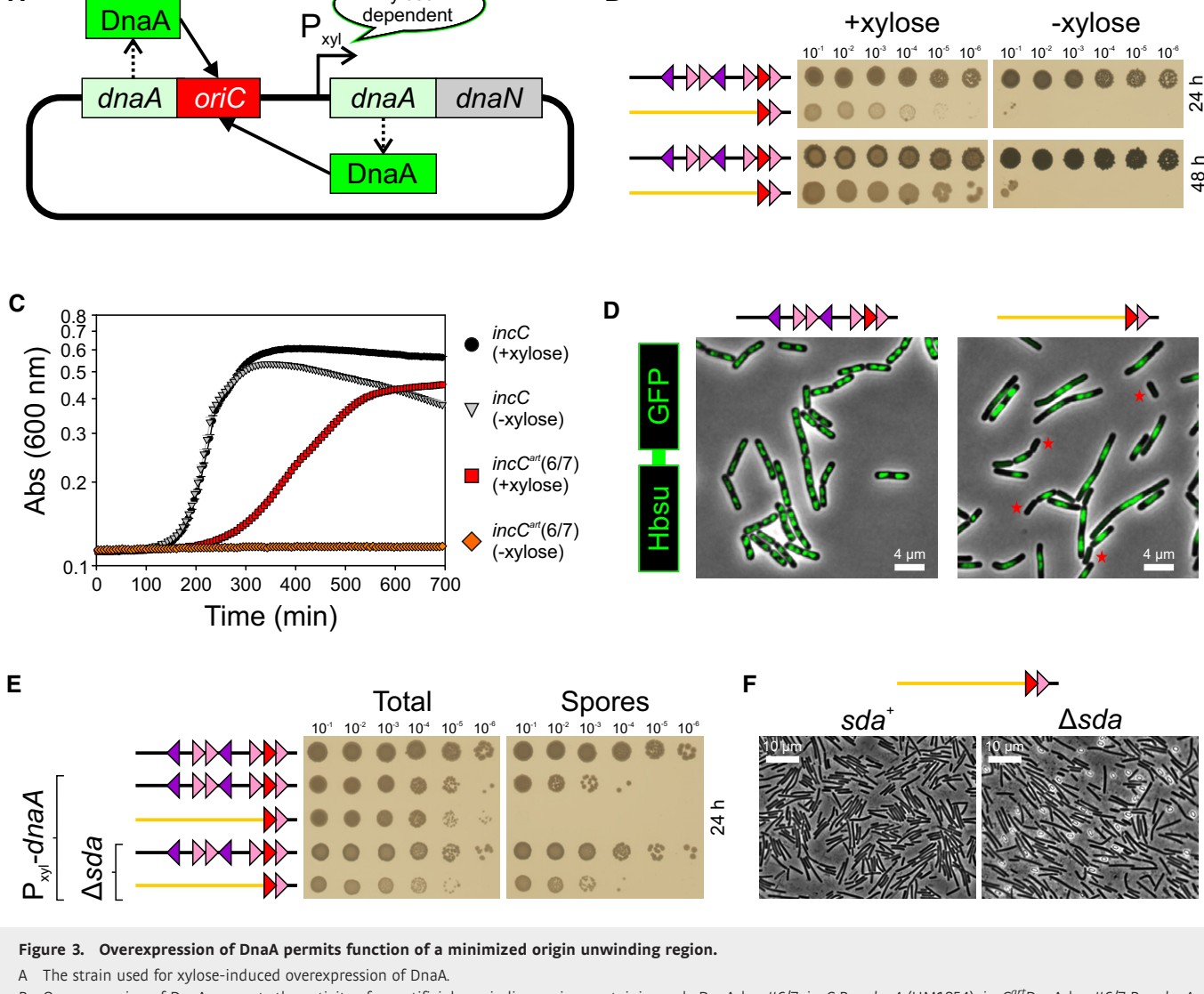

**Figure 3.  Overexpression of DnaA permits function of a minimized origin unwinding region.**

A   The strain used for xylose-induced overexpression of DnaA.
B   Overexpression of DnaA supports the activity of an artificial unwinding region containing only DnaA-box#6/7. *incC* $P_{xyl}$-*dnaA* (HM1854), *incC*$^{art}$DnaA-box#6/7 $P_{xyl}$-*dnaA* (HM1856).
C   The strain with a minimal artificial origin grows slower than wild-type in liquid media. *incC* $P_{xyl}$-*dnaA* (HM1854), *incC*$^{art}$DnaA-box#6/7 $P_{xyl}$-*dnaA* (HM1856).
D   Chromosome localization in live cells was determined using HBsu-GFP. Red stars indicate cells lacking DNA. *incC hbs-gfp* $P_{xyl}$-*dnaA* (HM1863), *incC*$^{art}$DnaA-box#6/7 *hbs-gfp* $P_{xyl}$-*dnaA* (HM1864).
E   Sporulation assay measuring the number of heat-resistant spores relative to total colony-forming units. *incC* (HM715), *incC* $P_{xyl}$-*dnaA* (HM1854), *incC*$^{art}$DnaA-box#6/7 $P_{xyl}$-*dnaA* (HM1856), *incC* $P_{xyl}$-*dnaA* Δ*sda* (HM1858), *incC*$^{art}$DnaA-box#6/7 $P_{xyl}$-*dnaA* Δ*sda* (HM1860).
F   Phase-contrast micrographs of minimal *incC* strains grown in sporulation medium with xylose. *incC*$^{art}$DnaA-box#6/7 $P_{xyl}$-*dnaA* (HM1856), *incC*$^{art}$DnaA-box#6/7 $P_{xyl}$-*dnaA* Δ*sda* (HM1860).

upstream of the native DnaA-box#6/7. In contrast to the wild-type DnaA chimera, all of the chimeric variants displayed severe growth defects (Fig 4D). To confirm that the DnaA chimeras were being expressed, each mutant was transformed into a Δ*dnaA* strain harbouring *oriN*; this allowed unambiguous detection of chimeric DnaA proteins using an anti-DnaA polyclonal antibody. Immunoblot analysis showed that the DnaA chimeras were stably expressed following induction with xylose (Fig 4E). These results suggest that activity of the DnaA protein being delivered from the upstream DnaA-box to the site of DNA unwinding requires amino acid residues known to participate in filament assembly and ssDNA binding.

## Reconstitution of DnaA-dependent unwinding of the minimal *incC*

To directly test whether DnaA-box#6/7 and the DnaA-trios constitute the minimal set of sequence elements required for DnaA to unwind the *B. subtilis* chromosome origin, we developed an *in vitro* system to detect DnaA-dependent DNA strand separation. Three oligonucleotides were annealed to form a scaffold that contained DnaA-box#6/7 and the DnaA-trios (Fig 5A). The oligonucleotide complementary to the DnaA-trios was labelled with a fluorophore to detect its position within a native polyacrylamide gel. According to

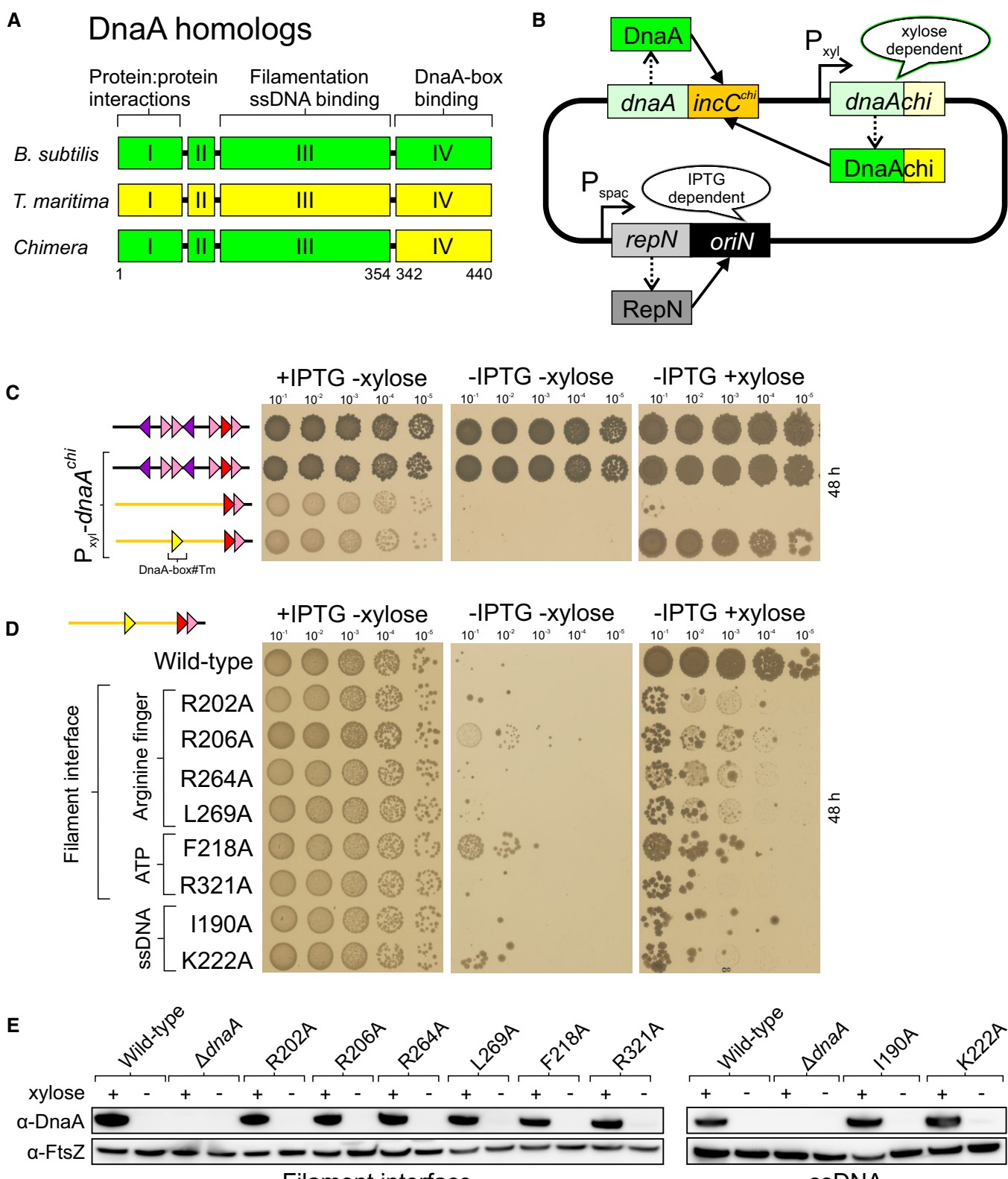

**Figure 4.**

the model, DnaA would initially bind to the DnaA-boxes using its dsDNA binding activity, followed by loading of an ATP-dependent DnaA filament onto the DnaA-trios (3′→5′) using its ssDNA binding activity. Specific engagement of the DnaA-trios would stretch the bound strand and destabilize hydrogen bonding with the complementary oligo, thereby liberating it from the scaffold. To perform

◀

**Figure 4.  DnaA targeted to the upstream DnaA-box must form a filament and bind ssDNA.**

A   Schematic diagram of the DnaA chimera.

B   The *oriC*-independent strain used for expressing chimeric DnaA proteins.

C   Activity of the chimeric DnaA protein requires a distal *T. maritima* DnaA-box. *incC* (HM1108), *incC* + P$_{xyl}$-*dnaAchi* (HM1683), *incC*$^{art}$DnaA-box#6/7 + P$_{xyl}$-*dnaAchi* (HM1694), *incC*$^{art}$DnaA-box#Tm$^{45}$/6/7 + P$_{xyl}$-*dnaAchi* (TR241).

D   Analysis of chimeric DnaA proteins localized at a distal *T. maritima* DnaA-box. DnaAchi (TR241), DnaAchi$^{R202A}$ (TR480), DnaAchi$^{R206A}$ (TR481), DnaAchi$^{R264A}$ (TR313), DnaAchi$^{L269A}$ (TR483), DnaAchi$^{F218A}$ (TR486), DnaAchi$^{R321A}$ (TR488), DnaAchi$^{I190A}$ (TR244), DnaAchi$^{K222A}$ (TR262).

E   Immunoblot analysis of the chimeric DnaA proteins in a Δ*dnaA oriN*$^{+}$ strain background. The tubulin homolog FtsZ was used as a loading control. DnaAchi (DS68), Δ*dnaA* (HM1423), DnaAchi$^{R202A}$ (DS61), DnaAchi$^{R206A}$ (DS62), DnaAchi$^{R264A}$ (DS60), DnaAchi$^{L269A}$ (DS64), DnaAchi$^{F218A}$ (DS65), DnaAchi$^{R321A}$ (DS66), DnaAchi$^{I190A}$ (DS57), DnaAchi$^{K222A}$ (DS58).

the unwinding assay as a function of time, a stop solution containing an excess of an unlabelled competitor oligonucleotide was used to inhibit reannealing of the fluorescently labelled probe (Fig 5B). Figure 5C shows that DnaA was able to separate the DNA strands in an ATP-dependent manner that also required ssDNA binding residue isoleucine 190.

To verify that DnaA filaments were forming during the unwinding reaction, a previously described crosslinking assay was employed (Scholefield *et al*, 2012). Here, two cysteine residues are introduced within the AAA + domain (DnaA$^{CC}$) such that when the DnaA filament assembles the cysteine residues from interacting protomers approach each other, allowing them to be captured by the cysteine-specific crosslinker bis(maleimido)ethane (BMOE; 8 Å spacer arm; Fig 5D). We confirmed that the DnaA$^{CC}$ variant remains capable of unwinding the wild-type DNA scaffold (Appendix Fig S5). Following crosslinking, DnaA$^{CC}$ oligomers larger than dimers were observed only in the presence of ATP and here they required isoleucine 190, indicating that DnaA$^{CC}$ assembles into a filament on a single DNA strand during the reaction (Fig 5E).

To confirm that the unwinding defect of the DnaA$^{I190A}$ mutant was specifically due to loss of ssDNA binding activity, an electrophoretic mobility shift assay was performed. Here, in addition to the Cy5-labelled oligo complementary to the DnaA-trios, the upstream oligonucleotide that forms part of the DnaA-boxes was labelled with Cy3 so that the position of both probes could be simultaneously detected (Fig 5F). Wild-type DnaA in the presence of ADP binds specifically to the DnaA-boxes (Fig 6D) and generates a single-shifted species that is observed for both the Cy5 and Cy3 probes (Fig 5G). Wild-type DnaA in the presence of ATP behaves differently, and the two fluorescent oligonucleotides no longer fully colocalize. Here, the Cy5-labelled probe was observed both as a larger species towards the top of the gel and below the unbound DNA scaffolds (Fig 5G). This is consistent with DnaA$^{ATP}$ assembling into filaments to generate the high molecular weight species and subsequently liberating a significant amount of the strand complementing the DnaA-trios (Fig 5G). In contrast, the Cy3-labelled oligonucleotide (DnaA-boxes) was only observed as the higher order species and was not separated from the complex, showing that DnaA unwinds the scaffold only at the DnaA-trios. The DnaA$^{I190A}$ variant in the presence of ADP was able to shift the scaffold similar to wild-type DnaA; however, in the presence of ATP it was unable to liberate the Cy5-labelled probe complementing the DnaA-trios and a smaller proportion of the Cy3-labelled probe was observed as high molecular weight species at the top of the gel (Fig 5G). These results indicate that the DnaA$^{I190A}$ variant can bind DnaA-boxes and adenosine nucleotides (i.e. note the differences between ADP and ATP in the electrophoretic mobility

shift assay), but that the protein is specifically defective in filament formation and DNA strand separation. Thus, it appears that DnaA must assemble into an ATP-dependent filament capable of engaging a single DNA strand in order to promote DNA unwinding activity.

### Unwinding the minimal *incC* requires DnaA-box#6/7 and DnaA-trios

To determine whether DnaA-box#6/7 and DnaA-trios are necessary and sufficient for DnaA unwinding activity, scaffolds were assembled with each of these sequences scrambled. In the case of the DnaA-trios, the base composition of the mutant scaffold was maintained to ensure that the stability of the DNA duplex was equivalent to wild-type. Mutating either the DnaA-boxes or the DnaA-trios severely inhibited the unwinding activity of DnaA (Fig 6A and B). An electrophoretic mobility shift assay showed that mutating the DnaA-boxes abolished DnaA binding to the scaffold (Fig 6C and D), and a BMOE crosslinking assay confirmed that mutating the DnaA-trios abrogated DnaA$^{CC}$ filament formation (Fig 6E and F). Finally, to explore whether the DnaA-trio motifs were acting as trinucleotide ssDNA binding elements, DNA unwinding was analysed using mutated scaffolds with either a single base pair or three base pair deletion. A single base pair deletion would be expected both to disrupt one DnaA-trio site and to shift the register of downstream DnaA-trios relative to the DnaA-boxes, whereas a three base pair deletion would be expected only to disrupt one DnaA-trio site (i.e. the downstream sites would be shifted but remain in register) (Fig 6G). As predicted for a trinucleotide motif, deletion of a single base pair (ΔA) significantly slowed the rate of DNA strand separation, while the additional deletion of the two adjacent base pairs (ΔATG) restored the unwinding rate to a level comparable with the wild-type substrate (Fig 6H).

### DnaA-box#6/7 loads DnaA filaments onto DnaA-trios *in cis*

The proximity of the specific DNA sequences within the *B. subtilis* unwinding region suggested a mechanism whereby DnaA proteins bound to DnaA-box#6/7 guide the formation of a DnaA filament onto the adjacent DnaA-trios. However, it has been proposed that during replication initiation in *E. coli,* DnaA binds DnaA-boxes upstream of the unwinding site and engages ssDNA that is delivered by the bending protein IHF, thereby simultaneously contacting dsDNA and ssDNA (Ozaki & Katayama, 2012a; Noguchi *et al*, 2015; Shimizu *et al*, 2016; Sakiyama *et al*, 2017). Although the minimal artificial unwinding sequence of *B. subtilis* used *in vivo* did not contain specific DnaA binding sites upstream of DnaA-box#6/7, it

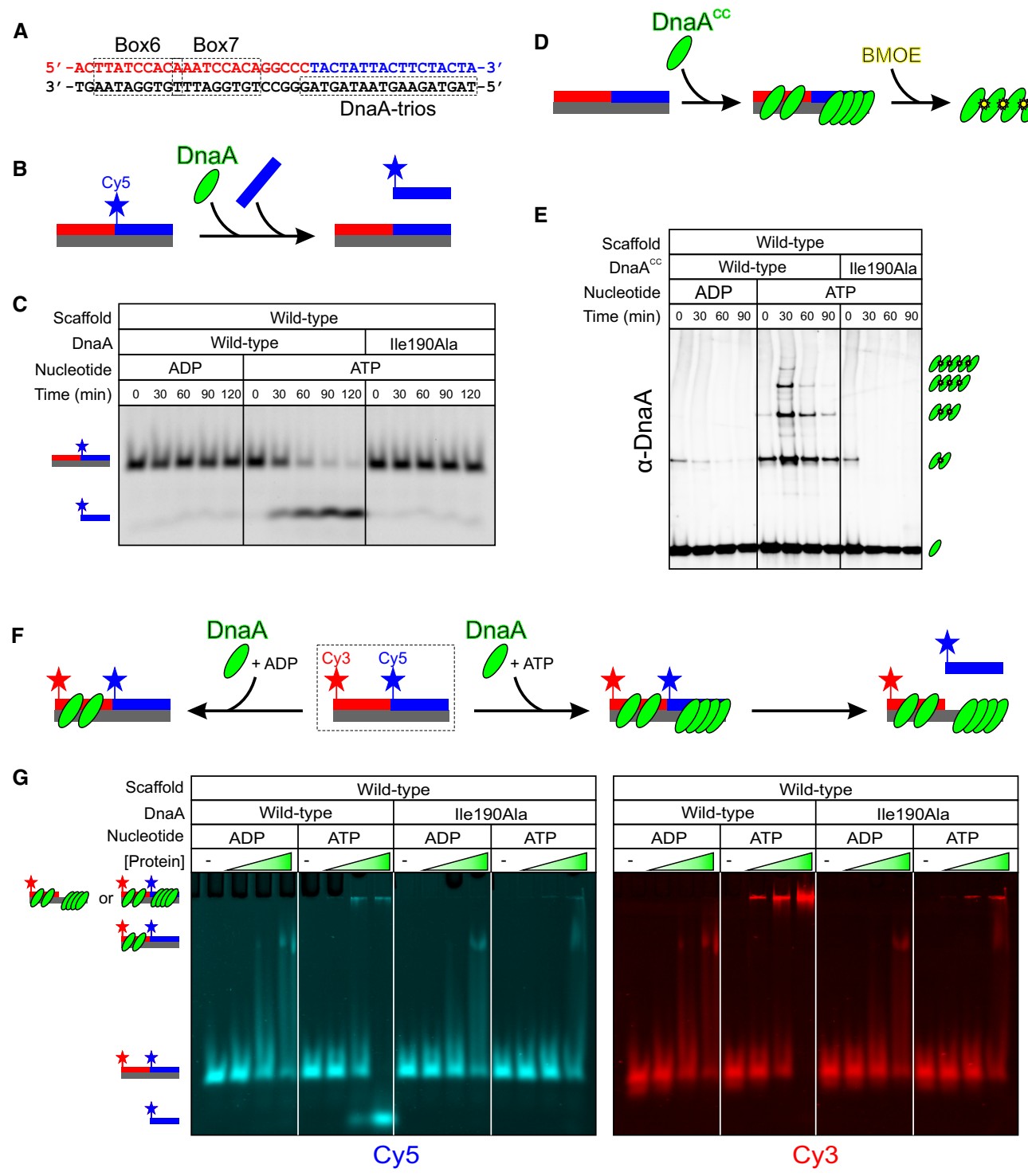

**Figure 5. Unwinding of the minimal *incC* requires DnaA filament assembly and ssDNA binding activity.**

A Sequences of the three oligonucleotides (coloured) used to construct a wild-type DNA scaffold. DnaA-box#6/7 and DnaA-trios are indicated.
B Schematic of the unwinding assay using a DNA scaffold and a competitor oligonucleotide.
C DnaA-dependent strand separation requires ATP and ssDNA binding activity.
D Schematic of the DnaA^CC filament assembly assay using a DNA scaffold and the cysteine-specific crosslinker BMOE.
E DnaA^CC filament assembly on the DNA scaffold requires ATP and ssDNA binding activity.
F Schematic of the electrophoretic mobility shift assay (EMSA) using a double-labelled wild-type DNA scaffold.
G EMSA showing that both wild-type DnaA and DnaA^I190A proteins bind the scaffold (+ADP) and that only the oligonucleotide complementary to the DnaA-trios is separated from the scaffold in the presence of ATP. Protein concentration was 0, 63, 125, 250 nM.

remained possible that DnaA could bind upstream through either formation of a DnaA filament on dsDNA emanating from DnaA-box#6 (Erzberger *et al*, 2006; Scholefield *et al*, 2012) or stabilization by a partner DNA binding protein (Chodavarapu *et al*, 2008). Moreover, because the *in vitro* unwinding assay used DNA scaffolds free in solution, these experiments would permit DnaA to bind two DNA molecules at once, potentially engaging ssDNA *in trans*. To test these alternative models (Figure 7A), we designed experiments aimed at determining whether DnaA binds DnaA-trios and promotes DNA strand separation *in cis* or *in trans*.

First, we combined in a single reaction scaffolds containing wild-type DnaA-boxes and mutated DnaA-trios with scaffolds containing mutated DnaA-boxes and wild-type DnaA-trios (Fig 7B). If DnaA engages DnaA-trios *in trans*, then the scaffold containing mutated DnaA-boxes and wild-type DnaA-trios should be a suitable substrate for strand separation. However, under these conditions we did not observe DNA unwinding, suggesting that DnaA is not acting *in trans* (Fig 7C).

Second, we captured biotin-labelled DNA substrates on streptavidin-coated magnetic beads to significantly decrease the effective concentration of DNA scaffolds in the reaction (> 100-fold) and therefore limit the potential for unwinding *in trans* (Fig 7D and Appendix Supplementary Methods). The rate of unwinding, as judged by liberation of the fluorescently labelled oligo complementing the DnaA-trios, was similar between scaffolds with or without biotin (Fig 7E). This result indicates that DNA unwinding occurs *in cis* with DnaA loaded from DnaA-boxes onto the adjacent DnaA-trios. The observed DNA unwinding of the captured scaffolds was ATP-dependent and significantly inhibited by substituting isoleucine 190 with alanine (Appendix Fig S6), confirming that the reaction proceeds via DnaA filament formation and ssDNA binding.

## Discussion

### Identification of a basal bacterial chromosome origin unwinding system

Despite extensive study, it remained unclear how the information encoded within a bacterial chromosome origin instructs DnaA to unwind the DNA duplex. To address this fundamental question, we employed the genetically tractable model organism *B. subtilis* and determined the precise sequences within the *incC* unwinding region that were necessary and sufficient for origin activity *in vivo*. This analysis indicated that DnaA binding sites are required at two

subregions containing DnaA-boxes, one proximal and one distal to the DnaA-trios. Characterization of the distal subregion suggests that it acts to increase the local concentration of DnaA at the site of origin unwinding.

The *in vivo* analysis also resulted in the conception of a highly minimized functional unwinding module containing only DnaA-box#6/7 adjacent to the DnaA-trios. Based on these findings, it became possible to reconstitute an origin-specific DnaA-dependent unwinding reaction *in vitro* using just these core *incC* sequences. The results are consistent with a mechanism in which DnaA$^{ATP}$ filaments are guided from the DnaA-boxes onto the adjacent DnaA-trios where they stretch ssDNA to promote origin unwinding (Fig 7F). We propose that this system constitutes a basal bacterial chromosome unwinding mechanism.

Following initial origin melting by DnaA, binding of single-strand binding protein can enlarge the unwound region further downstream (Krause & Messer, 1999). In this scenario, the location of a DnaA$^{ATP}$ filament on the unpaired DnaA-trios would position it to act as a docking site for a AAA+ helicase loader (Mott *et al*, 2008), guiding it onto ssDNA for recruitment and deposition of the replicative helicase.

### The basal unwinding system as a building block for bacterial chromosome origins

Founded on the following considerations, we propose that the basal architecture described here could be the ancestral bacterial chromosome origin unwinding module. First, the arrangement of a DnaA-box adjacent to DnaA-trios is conserved throughout the domain *Bacteria* (Richardson *et al*, 2016). Second, *B. subtilis oriC* is located adjacent to *dnaA,* as observed for the majority of bacterial chromosome origins (Mackiewicz *et al*, 2004; Luo *et al*, 2018). Third, *B. subtilis* is a member of the phylum *Firmicutes,* which is proposed to have diverged from other bacterial phyla at a relatively early age (Ciccarelli *et al*, 2006; Lake *et al*, 2009). Fourth, the basal origin unwinding system supports *B. subtilis* endosporulation (Fig 3E and F), a developmental process which likely evolved ~3 billion years ago (Battistuzzi *et al*, 2004).

Beginning with the system of a DnaA-box and DnaA-trios, new DNA binding sites for DnaA or other regulatory proteins could be superimposed onto these elementary sequence elements to adapt mechanical or regulatory inputs in different species. In *B. subtilis,* the addition of a DnaA-box upstream of the basal unwinding system allows for a lower functional level of DnaA within the cell (Figs EV1 and EV4). Duplications of upstream sites endow the system with robustness to mutation, as well as enabling optimal origin efficiency

**Figure 6.  DnaA-boxes and DnaA-trios are required for specific DNA unwinding by DnaA.**

A    Schematic of the DNA unwinding assay using an oligonucleotide scaffold.

B    DNA unwinding requires DnaA-boxes and DnaA-trios.

C    Schematic of the electrophoretic mobility shift assay using an oligonucleotide scaffold.

D    Electrophoretic mobility shift assay showing that DnaA-boxes are required to form a stable complex with DnaA. ADP was present at 1 mM, and DnaA concentration was 0, 63, 125, 250 nM.

E    Schematic of the DnaA$^{CC}$ filament assembly assay using a DNA scaffold and the cysteine-specific crosslinker BMOE.

F    DnaA$^{CC}$ filament assembly on the DNA scaffold requires DnaA-trios.

G    Schematic of the DNA unwinding assay using oligonucleotide scaffolds with either one (ΔA) or three (ΔATG) base pair deletions.

H    The rate of DNA unwinding decreases when one base pair is deleted but is similar to wild-type when three base pairs are deleted.

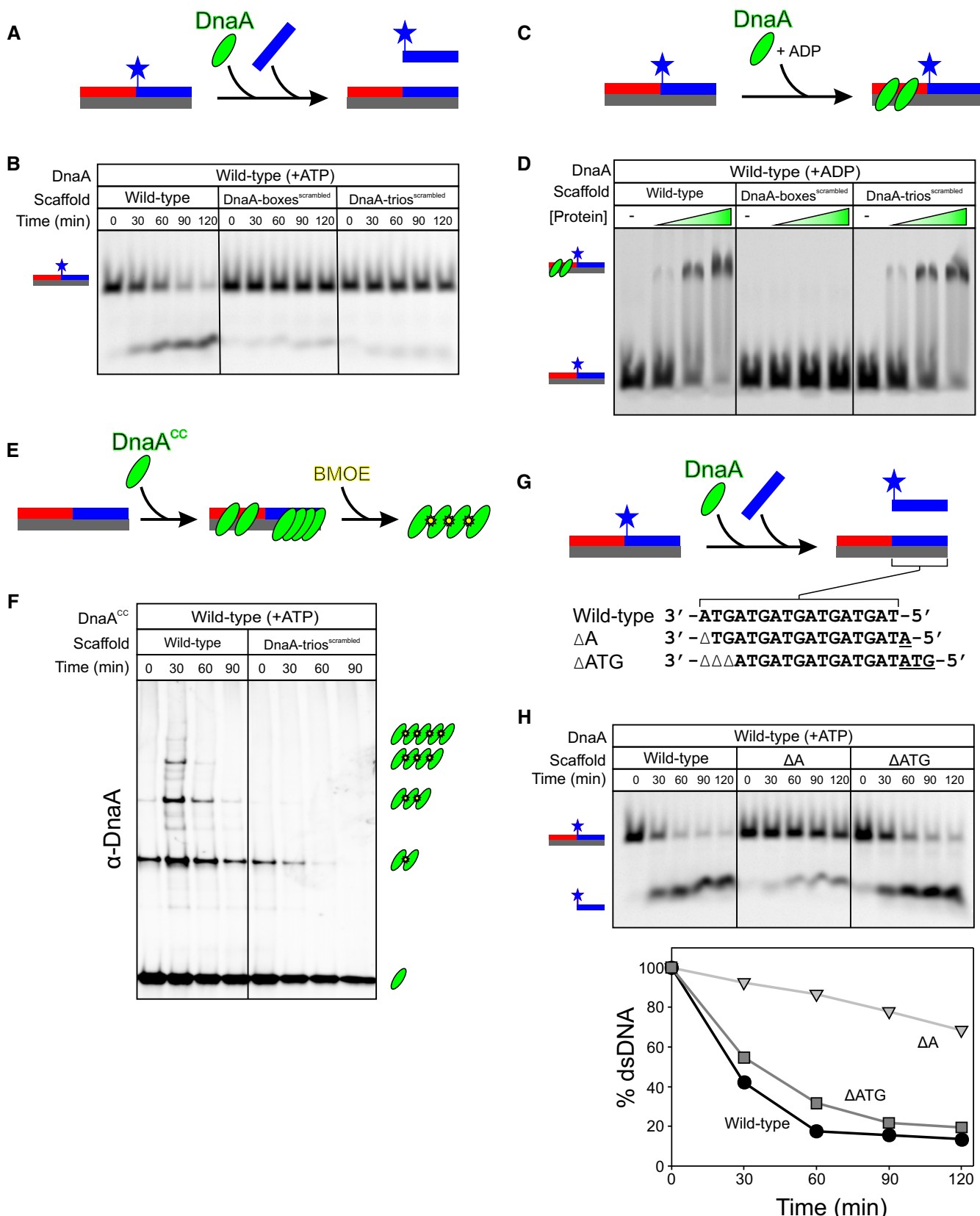

**Figure 6.**

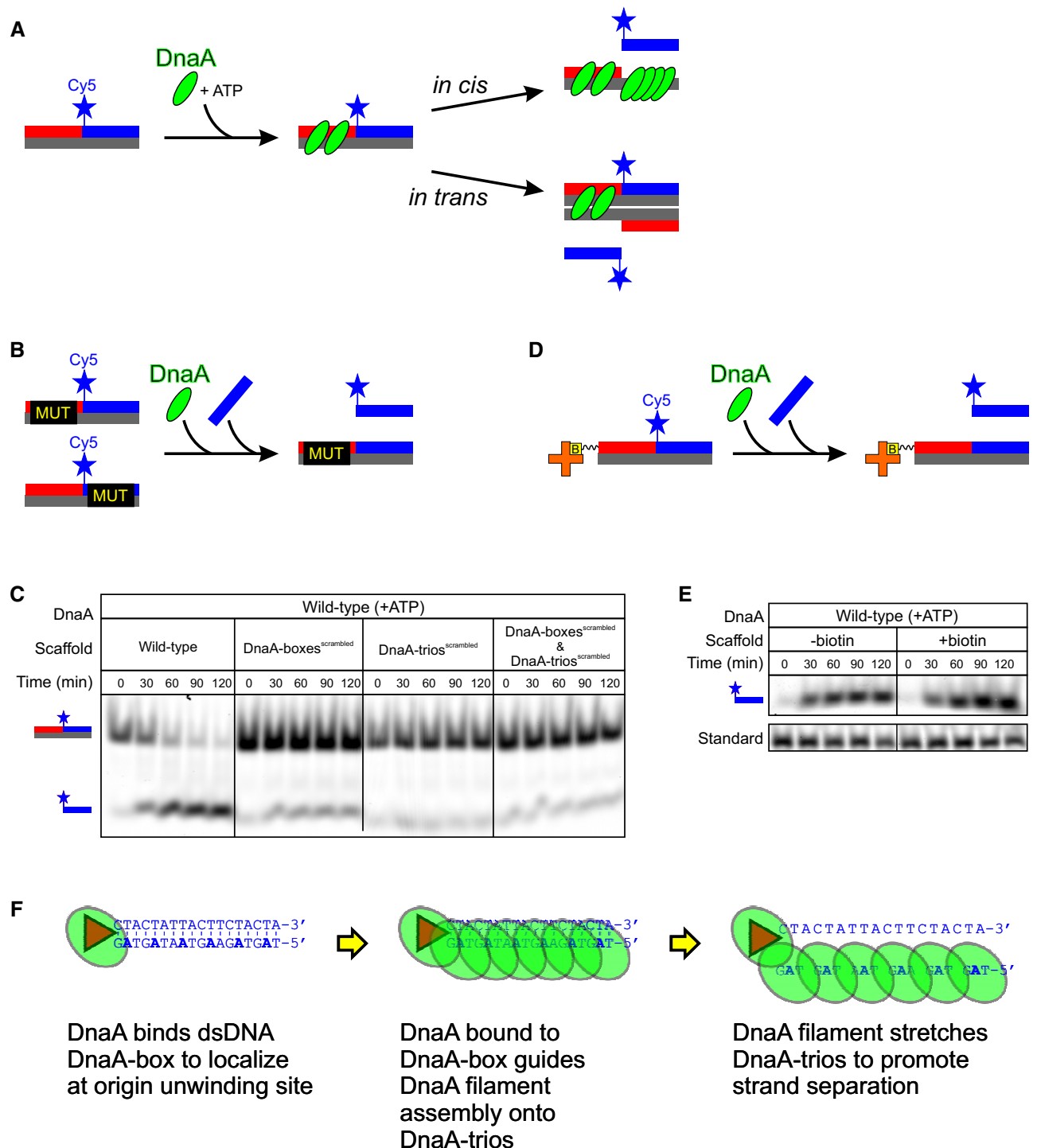

**Figure 7. DnaA filaments are loaded from DnaA-boxes onto DnaA-trios *in cis*.**

A  Alternative models illustrating how DnaA could engage DnaA-trios either *in cis* or *in trans*.

B  Schematic of a mixing experiment between DNA scaffolds with mutations in either the DnaA-boxes or DnaA-trios. Unwinding *in trans* is depicted.

C  DnaA cannot unwind a mixture of DNA scaffolds with mutations in either the DnaA-boxes or DnaA-trios. The total concentration of DNA scaffolds in all reactions was 25 nM.

D  Schematic of an experiment using biotinylated DNA scaffolds captured on streptavidin-coated beads.

E  DnaA unwinds DNA scaffolds that are captured on streptavidin-coated beads at the same rate as DNA scaffolds in solution.

F  The proposed basal bacterial chromosome unwinding system. DnaA uses its dsDNA binding activity to interact with a DnaA-box and subsequently guides the assembly of an ATP-dependent DnaA filament onto the DnaA-trios through its ssDNA binding activity. The DnaA filament stretches the ssDNA to separate the DNA strands and promote origin unwinding.

(Figs 1F, 3B–D and EV4). Based on the results in *B. subtilis* where less than half of the seven DnaA-boxes within the unwinding region are essential, we speculate that the majority of DnaA-boxes within bacterial origins are either redundant or regulatory (i.e. coordinating the timing or synchrony of origin firing within the cell cycle). It will be interesting to determine whether other species utilize DnaA-boxes to increase the local concentration of DnaA at *oriC* and how other DNA sequences modify activity of the basal unwinding system in response to various inputs such as the cell cycle, growth rate and growth phase.

### Architecture and composition of the DNA loop within *incC*

Genetic analysis of the distal DnaA-box within *incC* showed that it displays helical phasing and requires a minimal distance away from the site of unwinding, consistent with formation of a loop between the upstream DnaA-box and the site of unwinding (DnaA-box#6/7 and/or DnaA-trios; Hochschild & Ptashne, 1986; Bellomy *et al*, 1988). It is intriguing to further consider the structure of the proposed DnaA loop, particularly in the light of the fact that DnaA binds to the DnaA-box as a monomer.

In classical models of DNA looping via transcription repressors such as LacI, the proteins assemble into tetramers that bind operator sites as dimers (Becker *et al*, 2013). This allows a single tetramer to simultaneously bind operators both distal and proximal to the regulated promoter, with binding to the proximal site alone mediating transcriptional repression (Oehler *et al*, 1990). Note in these cases that each LacI subunit within the tetramer binds dsDNA through its helix-turn-helix motif.

In contrast to LacI, the experiments here using the DnaA chimera suggest that the replication initiator bound at the distal DnaA-box requires ssDNA binding activity. In the simplest case, a single DnaA protein bound to the upstream DnaA-box would loop and directly engage a DnaA-trio (Fig EV5i). However, since $DnaA^{ATP}$ can cooperatively assemble into an oligomer on dsDNA surrounding a DnaA-box (Speck *et al*, 1999; McGarry *et al*, 2004; Erzberger *et al*, 2006), it is conceivable that a DnaA filament is being delivered to the DnaA-trios (Fig EV5ii). Such a DnaA oligomer would contain multiple ssDNA binding motifs with the potential to interact with the repeating DnaA-trio elements.

In a more elaborate scenario, DnaA could be delivered from the upstream DnaA-box onto the DnaA-trios through a protein: protein intermediate. For example, the DnaA protein bound to DnaA-box#6/7 could contact the looping DnaA protein through its $AAA^+$ interface as part of a growing filament. There is also evidence for DnaA oligomerization through domain I, either by forming a homodimer (Simmons *et al*, 2003) (Fig EV5iii) or by assembling into a multimer built upon a protein scaffold (e.g. HobA in *H. pylori* and likely DiaA in *E. coli*; Natrajan *et al*, 2007, 2009). Interestingly, electron microscopy shows that HobA promotes DnaA-dependent bridging of two separate DNA molecules (Zawilak-Pawlik *et al*, 2007). In *B. subtilis*, the tetrameric replication initiation protein DnaD (Huang *et al*, 2008; Schneider *et al*, 2008) binds to domain I of DnaA (Matthews & Simmons, 2018; Martin *et al*, 2019), making it an attractive candidate to facilitate DnaA oligomerization (Fig EV5iv). Such multimeric intermediates would likely be relatively flexible complexes, which is consistent with the observation that the DnaA-box at the distal

*incC* subregion can function in either orientation and at great distances away from the DnaA-trios.

### Bacterial chromosome origin diversification

Although the blueprint of a DnaA-box adjacent to DnaA-trios is found throughout the bacterial domain, it is conspicuous that not all bacterial chromosome origins appear to harbour this arrangement. For example, the *E. coli* chromosome origin has undergone a major rearrangement and is no longer located proximal to the *dnaA* gene, consistent with the notion that it has diverged from the ancestral system. Although the minimal set of essential sequences at *E. coli oriC* has not yet been determined and the mechanism of strand separation is not clear, there is evidence that the replication initiation mechanism involves a DNA loop where DnaA binding at upstream double-strand DnaA-boxes (R5M/τ2/I1/I2) captures a single DNA strand from the downstream unwinding region (13-mers; Ozaki & Katayama, 2012a; Noguchi *et al*, 2015; Sakiyama *et al*, 2017). We propose that such a situation could have arisen from the basal system described here for *B. subtilis*. In this scenario, the initial regulatory role of the upstream DnaA-boxes to deliver DnaA from dsDNA onto ssDNA deviated to generate the inverse situation where the DNA loop delivers ssDNA to DnaA bound at the upstream DnaA-boxes (Fig EV4). Thus, understanding the basal *oriC* unwinding system in *B. subtilis* will guide identification of the principal sequences that are shared amongst seemingly diverse bacterial chromosome origins and provide a framework for considering regulatory and mechanistic alternatives.

# Materials and Methods

### Media and chemicals

Nutrient agar (NA, Oxoid) was used for routine selection and maintenance of both *B. subtilis* and *E. coli* strains. For experiments, *B. subtilis* cells were grown using Luria–Bertani (LB) medium, modified Schaeffer's media (0.8% Difco Nutrient Broth, 0.1% KCl, 1 mM MgSO$_4$, 1 mM CaCl$_2$, 0.1 mM MnSO$_4$) or Spizizen salts (1.4% K$_2$HPO$_4$, 0.6% KH$_2$PO$_4$, 0.2% (NH$_4$)$_2$SO$_4$, 0.1% Na$_3$-citrate·2H$_2$O, 0.02% MgSO$_4$·7H$_2$O). Supplements were added as required: 1% xylose, 0.1 mM IPTG, 100 μg/ml ampicillin, 5 μg/ml or 34 μg/ml chloramphenicol (for *B. subtilis* or *E. coli*, respectively), 1 μg/ml erythromycin, 5 μg/ml kanamycin and 50 μg/ml spectinomycin. Unless otherwise stated, all chemicals and reagents were obtained from Sigma-Aldrich.

### Phenotype analysis of *oriC* mutants

For analysis using solid media, strains were grown at 37°C in liquid cultures overnight to saturation in either LB or transformation medium supplemented with either IPTG (for P$_{spac}$-*oriN*) or xylose (for P$_{xyl}$-*dnaA*-*dnaN*). Serial dilutions were made, and 5 μl aliquots were spotted onto either NA plates or 1.5% agar plates containing Spizizen salts, 1 μg/ml Fe-NH$_4$-citrate, 6 mM MgSO$_4$, 0.5% glucose, 0.02 mg/ml tryptophan and 0.1% glutamate, with or without IPTG/ xylose. Plates were incubated at 37°C (see individual figure panels for time period). All experiments were independently performed at least twice, and representative data are shown.

For analysis using liquid media, strains were grown at 37°C in liquid cultures overnight to saturation in transformation medium supplemented with xylose (1%). Cells were diluted 1,000-fold into 100 µl LB with or without xylose (1%) in a 96-well plate (FALCON 353072) and covered with a gas-permeable membrane (Sigma Breathe-Easy Z380059). Plates were shaken (inside mode, high intensity, 1-s settling time) at 37°C in a Tecan Sunrise plate reader, and absorbance (600 nm, normal read mode) was measured every 6 min. Data were collected using Magellan software (version 7.2), exported to Microsoft Excel for analysis, and graphs were created using SigmaPlot (version 13.0).

### Immunoblot analysis of cell lysates

Whole cell extracts were prepared from cultures that were harvested at mid-log, resuspended in PBS + cOmplete EDTA-free protease inhibitor cocktail (Sigma-Aldrich) and sonicated. Proteins were separated by electrophoresis using a NuPAGE 4–12% Bis-Tris gradient gel run in MES buffer (Life Technologies) and transferred to a Hybond-P 0.45 µm PVDF membrane (GE Healthcare) using a semi-dry apparatus (Hoefer Scientific Instruments or Bio-Rad Trans-Blot Turbo). Proteins of interest were probed with polyclonal primary antibodies (Eurogentec) and then detected with an anti-rabbit horseradish peroxidase-linked secondary antibody produced in goat (Sigma) using an ImageQuant LAS 4000 mini digital imaging system (GE Healthcare). All experiments were independently performed at least twice, and representative data are shown.

### Sporulation assays

Strains were grown at 37°C in liquid cultures overnight to saturation in minimal transformation medium supplemented with xylose, then diluted 1,000-fold into modified Schaeffer's medium supplemented with xylose and grown at 37°C for 48 h. Sporulation frequencies were determined as the ratio of heat-resistant (80°C for 20 min) colony-forming units to total colony-forming units using serial dilutions and spotting 5 µl aliquots onto NA plates containing xylose. Plates were incubated at 37°C for 24 h. All experiments were independently performed at least twice, and representative data are shown.

### Microscopy

To visualize cells during the exponential growth phase, starter cultures were grown overnight in transformation medium supplemented with xylose and then diluted 1:100 into transformation medium supplemented with xylose, 0.1% glutamate and 0.2% casein hydrolysate. Cultures were allowed to achieve at least three doublings before observation. To visualize spores, strains were grown in Schaeffer's medium supplemented with xylose and grown at 37°C for 48 h.

Cells were mounted on ~1.4% agar pads (in sterile ultrapure water) and a 0.13- to 0.17-mm glass coverslip (VWR) was placed on top. Microscopy was performed on an inverted epifluorescence microscope (Nikon Ti) fitted with a Plan Apochromat Objective (Nikon DM 100x/1.40 Oil Ph3). Light was transmitted from a 300 Watt xenon arc lamp through a liquid light guide (Sutter Instruments), and images were collected using a Prime CMOS camera (Photometrics). The GFP

filter set was from Chroma: ET470/40x (EM), T495Ipxr (BS) and ET525/50m (EM). Digital images were acquired using METAMORPH software (version 7.7) and analysed using Fiji software (Schindelin *et al*, 2012). All experiments were independently performed at least twice, and representative data are shown.

### DNA scaffolds

DNA scaffolds were prepared adding each oligonucleotide (10 µM) in a 20 µl volume containing 30 mM HEPES-KOH (pH 8), 100 mM potassium acetate and 5 mM magnesium acetate. Mixed oligonucleotides were heated in a PCR machine to 95°C for 5 min and then cooled 1°C/min to 20°C before being stored at 4°C. Assembled scaffolds were diluted to 1 µM and stored at −20°C.

DNA scaffolds were imaged with a Typhoon FLA 9500 laser scanner (GE Healthcare). Cy5-labelled oligos were excited at 400 V with excitation at 635 nm and emission filter LPR (665LP). Cy3-labelled oligos were excited at 400 V with excitation at 532 nm and emission filter LPG (575LP). Images were processed using Fiji (https://doi.org/10.1038/nmeth.2019). All experiments were independently performed at least thrice, and representative data are shown.

### DNA unwinding assay

For standard DNA duplex unwinding assays, scaffolds (25 nM) were mixed with DnaA proteins (250 nM) and 1 mM nucleotide (ADP or ATP) in unwinding buffer containing 10 mM HEPES-KOH (pH 8), 100 mM potassium glutamate, 1 mM magnesium acetate, 30% glycerol and 10% DMSO. Reactions were incubated at 20°C, and at indicated times, 20 µl aliquots were removed and added to 3.7 µl stop solution containing 1,351 nM unlabelled competitor oligo, 2.2% SDS and 5.4 mg/ml proteinase K before being stored on ice.

When biotinylated scaffolds were used, 50 µl MyOne Streptavidin C1 Dynabeads (Thermo Fisher) were present for each sample. It was observed that 50% of the DNA was captured by the streptavidin-coated beads; thus, the scaffold concentration was 12.5 nM in these experiments. Reactions with biotinylated scaffolds were incubated at room temperature in a magnetic stand, and at indicated times, tubes were removed and mixed by gentle pipetting (to ensure samples were homogeneous, a non-specific fluorescently labelled oligonucleotide, 5′-Cy3-CTTTTTTCTTGGTCTCCCTGGGCACGTTCT-GATTCAACTGCTGAATCAGCTGCTC-3′, was used as a loading standard). 20 µl aliquots were taken and added to 3.3 µl stop solution containing 1,351 nM unlabelled competitor oligo and 5.4 mg/ml proteinase K before being stored on ice.

18 µl of each reaction was loaded onto a 6% polyacrylamide (19:1) gel and run at 100 V for 3 h in buffer containing 45 mM Tris, 45 mM boric acid and 5 mM magnesium chloride. All experiments were independently performed at least thrice, and representative data are shown.

### Electrophoretic mobility shift assay

For electrophoretic mobility shift assays, scaffolds (25 nM) were mixed with DnaA proteins (0, 63, 125, 250 nM) in 10 mM HEPES-KOH (pH 8), 100 mM potassium glutamate, 1 mM magnesium acetate, 30% glycerol, 10% DMSO and 1 mM nucleotide (ADP or ATP). Reactions (30 µl) were incubated at 20°C and after 2 h were

placed on ice for 5 min. 20 µl of each reaction was loaded onto a 6% polyacrylamide (19:1) gel and run at 70 V for 4 h in buffer containing 45 mM Tris, 45 mM boric acid and 5 mM magnesium chloride. All experiments were independently performed at least thrice, and representative data are shown.

**Filament assembly assay**

DnaA filament formation was promoted by mixing DnaA$^{CC}$ (250 nM) with DNA scaffold (25 nM) in 10 mM HEPES-KOH (pH 8), 100 mM potassium glutamate, 1 mM magnesium acetate, 30% glycerol, 10% DMSO and 1 mM nucleotide (ADP or ATP). Reactions were incubated at 20°C, and at indicated times, 10 µl aliquots were removed and crosslinked by addition of 2.5 µl of 20 mM bismaleimidoethane (BMOE; Thermo Fisher Scientific) (4 mM final). Reactions were incubated at 20°C for 6 min before quenching by addition of 5 µl of 200 mM cysteine (60 mM final) for 10 min. Samples were fixed in NuPAGE LDS sample buffer (Thermo Fisher Scientific) at 98°C for 5 min. Complexes were resolved by running 10 µl from each reaction on a NuPAGE Novex 3–8% Tris-Acetate Gel (Thermo Fisher Scientific), then transferred to PVDF membrane using Turbo-Blot transfer apparatus and Trans-Blot® Turbo™ Midi PVDF Transfer Packs (Bio-Rad). Complexes were visualized by immunoblotting using a polyclonal anti-DnaA antibody (Eurogentec). The interpretation of the crosslinked species is based on their migration relative to molecular weight markers. All experiments were independently performed at least twice, and representative data are shown.

**Protein purification**

Wild-type DnaA, DnaA$^{I190A}$, DnaA$^{CC}$ and DnaA$^{CC,I190A}$ were purified as follows. Proteins were expressed in BL21(DE3)-pLysS, grown to A$_{600}$ of 0.4 in LB medium at 37°C, then induced with 1 mM IPTG and cultured at 30°C for a further 3 h. Cells were harvested by centrifugation, resuspended in 45 ml of resuspension buffer (25 mM HEPES-KOH [pH7.6], 250 mM potassium glutamate, 10 mM magnesium acetate, 20% sucrose, 30 mM imidazole and 1 EDTA-free protease inhibitor tablet) and flash-frozen in liquid nitrogen. Cell pellet suspensions were thawed on ice with 32 mg of lysozyme and gentle agitation for 1 h, then disrupted by sonication at 20 W for 5 min in 2-s pulses. Cell debris was pelleted by centrifugation at 31,000 g, 4°C for 45 min, and the supernatant was further clarified by filtration (0.45 µm). All subsequent steps were performed at 4°C unless otherwise stated.

The clarified lysate was applied at 1 ml/min to a 1 ml HisTrap HP column (GE), which had previously been equilibrated with Ni binding buffer (25 mM HEPES-KOH [pH7.6], 250 mM potassium glutamate, 10 mM magnesium acetate, 20% sucrose and 30 mM imidazole). The loaded column was washed with a 10 ml 1 step gradient of 10% Ni elution buffer (25 mM HEPES-KOH [pH7.6], 250 mM potassium glutamate, 10 mM magnesium acetate, 20% sucrose and 30 mM imidazole). Specifically bound proteins were eluted using a 7.5 ml 1 step gradient of 100% Ni elution buffer and the entire fraction collected and diluted into 42.5 ml of Q binding buffer (30 mM Tris–HCl [pH7.6], 100 mM potassium glutamate, 10 mM magnesium acetate, 1 mM DTT and 20% sucrose). The diluted fraction was then applied at 1 ml/min to a 1 ml HiTrap Q HP column (GE), which had previously been equilibrated with Q binding buffer. The loaded HiTrap Q

HP column was washed with 10 ml of Q binding buffer, then eluted using a linear 10 ml gradient of 0–100% Q elution buffer (30 mM Tris–HCl [pH7.6], 1 M potassium glutamate, 10 mM magnesium acetate, 1 mM DTT and 20% sucrose) with 1 ml fractions collected. The peak 3 × 1 ml fractions, as judged by UV absorbance, were pooled and dialysed into 1 l of Factor Xa cleavage buffer (25 mM HEPES-KOH [pH7.6], 250 mM potassium glutamate, 20% sucrose and 5 mM CaCl$_2$), using 3.5k MWCO SnakeSkin dialysis tubing (Life Technologies) at 4°C overnight.

The dialysed protein was diluted to 5 ml total volume in Factor Xa cleavage buffer and incubated at 23°C for 6 h with 80 µg of Factor Xa protease (NEB). The sample was applied at 1 ml/min to a 1 ml HisTrap HP column (GE), which had previously been equilibrated with Factor Xa cleavage buffer. The Factor Xa-cleaved fraction was eluted in 7.5 ml of Ni binding buffer. The eluted fraction was diluted into 42.5 ml of Q binding buffer and purified on a 1 ml HiTrap Q HP column as previously described. Peak fraction(s) were pooled and dialysed into 1 l of final dialysis buffer (40 mM HEPES-KOH [pH7.6]; 250 mM potassium glutamate; 1 mM DTT; 20% sucrose; 20% PEG$_{300}$), using 3.5k MWCO SnakeSkin dialysis tubing (Life Technologies) at 4°C overnight before aliquoting, flash-freezing in liquid nitrogen and storage at -80°C. Removal of the N-terminal His-tag, following incubation with Factor Xa, was confirmed by anti-pentaHis (Qiagen) Western blotting.

**Expanded View** for this article is available online.

## Acknowledgements

We thank Waldemar Vollmer, Yoshikazu Kawai, Katarzyna Mickiewicz and Charles Winterhalter for critical review of the manuscript. We thank Alan Koh for Erasmus student supervision, Christopher Blower for help with plasmid construction and Frances Davison for technical assistance. Research support was provided to HM by a Royal Society University Research Fellowship (UF080009), a Wellcome Trust Senior Research Fellowship (204985/Z/16/Z) and grants from the Biotechnology and Biological Sciences Research Council (BB/K017527/1 and BB/P018432/1). DS was supported by a Research Excellence Academy Studentship from the Faculty of Medical Sciences at Newcastle University. OH was supported by an Iraqi Ministry of Higher Education and Scientific Research Studentship. TS was supported by the Erasmus+ programme of the European Union.

## Author contributions

TTR, DS, SP, OH, TS and HM generated results presented in the manuscript. TTR, DS, SP and HM created Figs. HM wrote the manuscript. TTR, DS and SP edited the manuscript.

## Conflict of interest

The authors declare that they do not have any conflicts of interest.

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
