## [Review Process File · The EMBO Journal]

Identification of a basal system for unwinding a bacterial chromosome origin

Tomas T. Richardson, Daniel Stevens, Simone Pellicciari, Omar Harran, Theodor Sperlea and Heath Murray.

Review timeline:

Submission date:	28 th January 2019
Editorial Decision:	27 th February 2019
Revision received:	23 rd April 2019
Editorial Decision:	5 th June 2019
Revision received:	6 th June 2019
Accepted:	7 th June 2019

Editor: Hartmut Vodermaier

Transaction Report:

1st Editorial Decision

27th February 2019

Thank you for submitting your manuscript on bacterial replication origin unwinding to The EMBO Journal. We have now received comments from three expert referees, copied below for your information. Given that all reviewers find the study well conducted and presented, and its findings interesting and potentially important, we would be happy to consider it further for publication, pending satisfactory addressing of a number of specific concerns raised in all three reports. As you will see, while many points pertain to aspects of presentation/interpretation/discussion, there are also several issues that will require further control experiments to rule out possible alternative explanations and scenarios.

Should you be able to adequately address these various points, we shall be happy to consider a revised manuscript further for publication in The EMBO Journal.

REFEREE REPORTS

Referee #1:

All the vital transactions on DNA require its strand to be open. It is still a challenge to understand how this energetically unfavorable reaction happens. The Murray lab in their 2016 Nature paper identified an array of trinucleotide repeats (Trios) and claimed that DnaA binds to those repeats in their single-stranded form. Single-stranded binding of DnaA was proposed from crystallographic evidence earlier but sequence specificity of the binding and the phasing of Trio repeats are clearly shown here (Fig 6G). The authors also show results that support the stretching of single-stranded DNA upon DnaA binding. The most laudatory aspect of the study is the demonstration of a minimal system for origin opening involving both double and single-stranded binding of DnaA. This shows that the authors are in the right track in their seemingly radical proposition of the mechanism for origin opening. The findings here can be considered as major advances in the field of DNA-protein interactions relevant to opening the strands of DNA. This work is rigorous, well controlled and well

written. I have no doubt that the work will encourage others to try to identify Trios and test their function in other bacteria.

How DnaA binding switches from double to single-stranded DNA upon encountering Trios is a major unknown in the field. The authors propose that the double-stranded binding helps to increase the local concentration of DnaA in the vicinity of Trios. The details of this aspect of the work is a subject for a separate study but the authors can clarify things a little better as I have elaborated in my comments #2&4 below. The other comments below are minor.

1. L.137: I would say origin activity rather than incC activity, which is defined as an inhibitory activity (l.120).
2. L.150-4: To argue for the requirement of both distal and proximal DnaA boxes, the authors could have tried ori with #5,6 and #5,6,7 only. If they conferred function, then proximal sites could have been enough. Remember, in *E. coli*, the (DUE) distal set of DnaA boxes can be deleted without losing origin function (Bates/ Katayama).
3. L.177: function.
4. L.182: "phasing and minimal distance"- the present results mimic what Hochschild found in lambda; OR1 and OR2 can either be side by side or at a minimum of four turns of the helix apart. The authors should be careful in concluding the requirement for a distal site. Try 567. If 567 does not work, then increasing local concentration may not be the (only) role of distal sites. The delivery by DnaA looping might be a requirement.
5. L.202: vector---was 'used' to transform---. (You transform the host, not the plasmid).
6. L.216: S5B would be better than S5.
7. L.238: Neither filament formation nor SS binding is shown here. The title and the inference (l.286-8) should be toned down, e.g., to "distal DnaA molecule contributes possibly by---".
8. L.249: should be required for.
9. Fig 4C: I think the figure would benefit from including the control that shows that DnaAchi effect is through incCchi. In other words, induce DnaAchi in a strain that do not have the incCchi and show that addition of xylose makes no difference.
10. L.278: expressed at.
11. Fig 7: It shows that DnaA is donated from double to strands. Is there any knowledge of DnaA binding affinity to the two forms of DNA? The model implies that Trios win at the expense double-stranded binding.
12. L.402-8: Remember opening is not the only goal of replication initiation. The origin needs to fire once and only once per cell cycle, and fire at a particular time of the cell cycle. It remains to be seen whether the basal system can measure up to these demands.
13. L.421: may say subunit instead of protein.
14. L.461: principal?

Referee #2:

Richardson and coworkers examined the DNA sequence elements in the replication origin of *Bacillus subtilis* that are necessary for its function. The work is carefully done, the conclusions are supported by the experimental results, and the manuscript is clearly written. Assuming that the authors are able to revise the manuscript satisfactorily, the work should appeal to those who study DNA replication and replication restart, and to the broad readership of The EMBO Journal. Specific issues are set out below.

Major (and not so major) comments:

1. Please consider revising the title. Although the contributions of individual DnaA boxes and DnaA-trio motifs in the replication origin of *Bacillus subtilis* have been analyzed to identify those that are necessary and sufficient for function in this organism, it is unclear whether the comparable sequences in the replication origins of other bacteria are also necessary and sufficient. The underlying problem with the title is the word "bacterial."
2. A reservation with the results is whether the minimal motifs of DnaA box 6 and 7 and the DnaA-trio region support synchronous initiation. Please consider adding a qualifying statement.
3. The publication by Duderstadt, Chuang and Berger (2011) suggests a model that ssDNA is

stabilized by interaction with amino acids in the interior of the DnaA filament. One experiment in the paper is a strand displacement assay in which the binding of DnaA to a short (15mer) dsDNA fluorescently-labeled in one strand led to partial displacement of the fluorescently-labeled ssDNA. A 30mer was inert. Strand displacement activity has been also been ascribed to T4 SSB, E. coli SSB, and hRPA that bind to ssDNA cooperatively. Other than DnaA boxes 6 and 7 serving as a nucleation site, is the "unwinding assay" measuring strand displacement?

4. Duderstadt, Chuang and Berger (2011) used a FRET-based assay to measure the binding of DnaA to dT-21, and suggested that DnaA stretches the ssDNA. The use of the term "stretch" is a little misleading. Compared with the flexibility of the free ssDNA, the increased distance measured by FRET analysis simply reflects the greater rigidity of the complex of DnaA bound to the ssDNA. Please consider using a different term.

5. Line 181 and 443. The 75 and 86 constructs are the only ones that work in both orientations. Please qualify the statements in the respective lines.

6. Line 204 and Figure S4. Isn't it possible that the increased abundance of DnaN is responsible? If so, please modify the statement in line 204.

7. Line 235-236. Isn't the data simply consistent with the requirement for a few rounds of DNA replication preceding the commitment to spore formation?

8. Figure 6F. In the lanes with DeltaDnaA-trios, the total amount of DnaA decreases with time. If DnaA is at the top of the gel and not visible in the panel, does this mean that the absence of DnaA-trios promotes concerted filament formation?

Minor comments:

9. Line 129 and elsewhere. The IPGT-regulated promoter is a little leaky. See Bhavsar et al., (AEM (2001) 67:403). Please revise the statement.

10. Line 153. DnaA box 2/3 + 6 shows partial activity. Please revise the text.

11. Figure 2A. Did the authors test the constructs with the DnaA-box#1 and 4 in the other orientation?

12. Figure 6F. Does the interpretation of the crosslinked species rely on molecular weight markers or are they deduced? If molecular weight markers were used, it may help to include their positions in Figure 6F.

13. Line 492-498 and elsewhere where serial dilutions are shown. Was the turbidity of each culture first adjusted so that they were equal before preparing the dilutions?

14. Figure 2B and 2D. What was the length of incubation?

15. Line 229. Isn't the decrease 100-fold (row 1 vs 2, Figure 3E)?

16. Line 276. F128A and E183A are not entirely wild type in activity.

17. Line 396. "a the"

18. Line 719. Do the authors want to also label the DnaA-trios region in Figure 1a with "incC unwinding region?"

19. Please consider removing Figure S1B.

Referee #3:

The manuscript by Richardson et al. considers the long-standing puzzle of what minimal sequence elements are required for replication origin function. They pursue a combined genetic and biochemical approach to evaluate the consequences of systematic changes to origin organization. The key technology used in vivo is an orthogonal replication initiation origin, oriN, that does not depend on the normal initiator DnaA. This tool, leveraged in earlier work from the same group, allows for the study of origin sequence elements without selecting for suppressor mutants. This strategy allowed the authors to identify a minimal origin sequence containing fewer DnaA boxes and revealed a critical spacing and phasing of binding sites. They further interrogate the mechanism of initiator engagement using a series of assembly and DNA binding mutants as well as a clever in vitro unwinding assay.

The manuscript is very well written with context provided by well-chosen references and clear figures. The strategies pursued are well matched with the central question considered. Moreover, the approach provides numerous novel insights into the organization of the *Bacillus subtilis* origin which, through comparative analysis, provide a framework for understanding the minimal conserved elements of all bacterial origins.

Several key models for DnaA mediated unwinding are discussed and the authors rightly point out that these models may not be mutually exclusive. The surprisingly huge distance tolerated between DnaA binding sites lends strong support to a loop-back mechanism, but the authors also demonstrate the critical importance of ssDNA binding consistent with a DNA stretching mechanism.

Overall, the experiments conducted were persuasive and support the conclusions drawn. However, several key conditions were not tested for a chimeric DnaA intended to evaluate the loop-back mechanism and in the *in vitro* unwinding assay (see major comments). The absence of these controls allows for several alternative models for the structural intermediates formed during origin unwinding. These alternatives were not mentioned or considered by the authors. A few other issues of clarity and remaining questions arose during the course of reviewing the manuscript that should be considered by the authors (minor comments).

Major Comments

1. The authors identified a minimal origin containing the closely spaced boxes#6/7 and a distal box#CR (box#3). Large distances between these elements were tolerated leading the authors to hypothesize that a loop is formed to allow the two clusters of DnaA oligomers to come together. To evaluate this possible loop interaction, the authors constructed a chimeric DnaA composed of the double-stranded binding domain from *Thermotoga maritima* with domains I-III of *Bacillus subtilis*. They then replaced the box#CR with a high affinity *Thermotoga maritima* binding site. They found that origin function then depended on expression of the chimeric DnaA and mutations in the chimera disrupting ssDNA binding and oligomerization disrupted origin function. Based on these findings, the authors concluded that there is a *trans* interaction supported by a loop in which the chimeric DnaA assembles with the *Bacillus subtilis* complex formed on boxes#6/7 and this involves oligomerization between the subcomplexes and the ssDNA binding activity of the chimera for origin unwinding.

There is an alternative interpretation consistent with these observations. Given that the primary oligomerization domain of the chimera is the same as for wild-type *Bacillus subtilis* DnaA, it is also possible a mixed complex forms on boxes#6/7 composed of wild-type *Bacillus subtilis* DnaAs bound to boxes#6/7 and chimeric DnaAs co-assembling through domain III. In this case, the combined expression of wild-type and chimera could be sufficient to support origin unwinding without the need for box#CR (box#3) as demonstrated by the authors for higher levels of wild-type expression alone.

To exclude this possibility the authors should do a control in which box#CR (box#3) is removed but both the chimera and the wild type are expressed and demonstrate this does not support origin unwinding. Furthermore, in Fig 4D the authors use a strain with an endogenous *dnaA* KO to confirm chimeric mutant expression. Is the wild-type *dnaA* viable in such a genetic background when overexpressed either in presence of only box6+7 (according to Fig 3B) or with the additional box3 T.m. (Fig 4C)?

A decisive experiment demonstrating the chimera and wild-type DnaA bind only the desired boxes hasn't been provided. The above experiments would clarify this issue. Alternatively, an EMSA could be used to confirm the specificity of the sites for the T.m. vs. wild-type DnaA variants.

In the absence of these controls, alternative binding models must be further discussed and would seem to alter the central interpretation put forward for the chimera.

2. Figure 5 and 6 show an *in vitro* unwinding assay on a short duplex substrate containing DnaA boxes#6/7 and the adjacent DnaA-trios. Addition of DnaA and a competing oligo for the trio region allows for unwinding. This authors show this activity depends on the DnaA binding sites and the trio. However, given the small size of the substrate it is hard to visualize how the adjacent DnaA molecules might engage with the trios.

An alternative explanation is that unwinding is conducted *in trans* wherein one substrate bound DnaA oligomer engages with the trio of another substrate. To test this possibility the authors should conduct an experiment in which they mix a substrate with functional DnaA binding sites and corrupted trio with a substrate with corrupted DnaA binding sites and a functional trio. If unwinding

still occurs in this case it would indicate complexes are acting in trans. Otherwise, unwinding occurs entirely within a single substrate as suggested in the manuscript.

Minor Comments

3. Fig 2B: Why were distance 86, 91 and 96 chosen for reversed box3 orientation? Does the helical phasing for the reversed orientation follow the same behavior as in Fig 2A?
4. Fig 2D: The DNA cartoon is fully aligned and looks the same scale as in Fig 2A & Fig 2B but the distances are much larger. Highlight the larger distance in a graphical way to clarify for the reader (e.g. by a scale break)
5. The trio elements should be introduced more fully in the introduction. Without knowledge of previous work from the group their importance for origin function is not entirely clear.
6. The sentence between lines 180 and 184 should be split or shortened and focused for clarity.
7. The authors mention the oligomerization activity of domain I in the discussion as an alternative mode for two distant DnaA subcomplexes to co-assemble. This could be tested directly by removing domain I from the chimeric DnaA and observing if origin function is still supported. Alternatively, perhaps the authors could point out some examples from the literature supporting this role for domain I.

1st Revision - authors' response

23rd April 2019

RESPONSE TO REVIEWERS

Referee #1:

1. L.137: I would say origin activity rather than *incC* activity, which is defined as an inhibitory activity (l.120).

This is a fair point. However, we believe that within its natural context the *incC* region is the site of unwinding and therefore this is the relevant activity to focus on. To make this point clear we have added the following statement: “note that throughout the present study we will refer to the activity of *incC* as supporting DNA unwinding within the endogenous origin” (line 127-129).

2. L.150-4: To argue for the requirement of both distal and proximal DnaA boxes, the authors could have tried ori with #5,6 and #5,6,7 only. If they conferred function, then proximal sites could have been enough. Remember, in *E. coli*, the (DUE) distal set of DnaA boxes can be deleted without losing origin function (Bates/ Katayama).

We created these mutants and find that they are non-functional, consistent with the model that both distal and proximal DnaA-boxes are required at *incC*. We have added this data to Appendix Fig S3A and described the results in the text (line 157-159).

3. L.177: function.

Changed.

4. L.182: "phasing and minimal distance"- the present results mimic what Hochschild found in lambda; OR1 and OR2 can either be side by side or at a minimum of four turns of the helix apart. The authors should be careful in concluding the requirement for a distal site. Try 567. If 567 does not work, then increasing local concentration may not be the (only) role of distal sites. The delivery by DnaA looping might be a requirement.

This is a good idea. However, during the course of this work we unexpectedly observed that introducing a consensus DnaA-box at position DnaA-box#5 inhibits the activity of an otherwise functional origin (e.g. DnaA-box#3/5^{con}/6/7). Our hypothesis is that the strong DnaA binding site located two base pairs upstream of DnaA-box#6 somehow impairs the function of the DnaA-box#6/7 module. We note that the two base pair spacing of DnaA-box#5 from DnaA-box#6 is ideal to promote cooperative binding between DnaA proteins (Rozgaja *et al.* 2011 Mol Micro). Thus, the interpretation of the proposed experiment would require additional efforts that we believe are beyond the scope of this study.

To address the Reviewer's concern we have removed the statement referring to the "minimal distance" and revised the text to say "activity displays specific phasing" rather than "activity requires specific phasing" (line 188-191).

5. L.202: vector----was 'used' to transform---. (You transform the host, not the plasmid).

Changed.

6. L.216: S5B would be better than S5.

Changed.

7. L.238: Neither filament formation nor SS binding is shown here. The title and the inference (1.286-8) should be toned down, e.g., to "distal DnaA molecule contributes possibly by---".

This is a fair point. We have changed the heading for this section to "Activity of the DnaA protein delivered from the upstream subregion requires residues involved in filament formation and ssDNA binding". We have also changed the concluding sentence to "These results suggest that activity of the DnaA protein being delivered from the upstream DnaA-box to the site of DNA unwinding requires amino acid residues known to participate in filament assembly and ssDNA binding" (line 300-302).

8. L.249: should be required for.

Changed to "would require".

9. Fig 4C: I think the figure would benefit from including the control that shows that DnaAchi effect is through incCchi. In other words, induce DnaAchi in a strain that do not have the incCchi and show that addition of xylose makes no difference.

This is a good point. We have now provided this critical experiment in Fig 4C and described the result in the text (line 280-282).

10. L.278: expressed at.

Changed.

11. Fig 7: It shows that DnaA is donated from double to strands. Is there any knowledge of DnaA binding affinity to the two forms of DNA? The model implies that Trios win at the expense double-stranded binding.

This is a good question. Using fluorescence polarization the *Aquifex aeolicus* DnaA was determined to have a K_d of ~100 nM for non-specific ssDNA (dT₂₅, dA₂₅, dC₂₅) while the *E. coli* DnaA was measured to have a K_d of ~50 nM for a consensus DnaA-box in dsDNA (Duderstadt *et al.* 2010 JBC). Assuming that DnaA will have a greater affinity for DnaA-trios compared to non-specific ssDNA, and allowing that additional protein:protein interactions could stabilize the interaction of DnaA with DnaA-trios (see Figure EV5), we believe that this model is reasonable. In future work we plan to characterize the role of the proposed DNA loop in greater detail and measurements of binding affinities for *Bacillus subtilis* DnaA on appropriate substrates will be included here.

12. L.402-8: Remember opening is not the only goal of replication initiation. The origin needs to fire once and only once per cell cycle, and fire at a particular time of the cell cycle. It remains to be seen whether the basal system can measure up to these demands.

We have modified this statement to say “Based on the results in *B. subtilis* where only two of the seven DnaA-boxes within the unwinding region are essential, we speculate that the majority of DnaA-boxes within bacterial origins are either redundant or regulatory (i.e. – coordinating the timing or synchrony of origin firing within the cell cycle)” (line 448-451).

13. L.421: may say subunit instead of protein.

Changed.

14. L.461: principal?

Changed.

Referee #2:

1. Please consider revising the title. Although the contributions of individual DnaA boxes and DnaA-trio motifs in the replication origin of *Bacillus subtilis* have been analyzed to identify those that are necessary and sufficient for function in this organism, it is unclear whether the comparable sequences in the replication origins of other bacteria are also necessary and sufficient. The underlying problem with the title is the word "bacterial."

We appreciate this point. However, we believe the combination of functional data from *B. subtilis*, combined with bioinformatic evidence showing that these basal elements are conserved in diverse bacterial origins (Richardson *et al.* 2016 Nature), make a logical case for proposing that the basal system is widespread throughout the domain. This viewpoint is reflected by Reviewer#3: "The approach provides numerous novel insights into the organization of the *Bacillus subtilis* origin which, through comparative analysis, provide a framework for understanding the minimal conserved elements of all bacterial origins."

2. A reservation with the results is whether the minimal motifs of DnaA box 6 and 7 and the DnaA-trio region support synchronous initiation. Please consider adding a qualifying statement.

We have added a qualifying statement to the Discussion: "Based on the results in *B. subtilis* where only two of the seven DnaA-boxes within the unwinding region are essential, we speculate that the majority of DnaA-boxes within bacterial origins are either redundant or regulatory (i.e. – coordinating the timing or synchrony of origin firing within the cell cycle)" (line 448-451).

3. The publication by Duderstadt, Chuang and Berger (2011) suggests a model that ssDNA is stabilized by interaction with amino acids in the interior of the DnaA filament. One experiment in the paper is a strand displacement assay in which the binding of DnaA to a short (15mer) dsDNA fluorescently-labeled in one strand led to partial displacement of the fluorescently-labeled ssDNA. A 30mer was inert. Strand displacement activity has been also been ascribed to T4 SSB, *E. coli* SSB, and hRPA that bind to ssDNA cooperatively. Other than DnaA boxes 6 and 7 serving as a nucleation site, is the "unwinding assay" measuring strand displacement?

This is an interesting question. The proposed mechanism for how DnaA engages and stretches a single DNA strand suggests that it contacts the surface of the DNA backbone, rather than the surface of the bases, such that it would not be positioned to physically displace the complementary strand (Duderstadt *et al.* 2011 Nature). Mutagenesis *in vitro* and *in vivo* supports the crystallographic model (Duderstadt *et al.* 2011 Nature; Ozaki *et al.* 2008 JBC; this manuscript). Taken together we believe the data supports the model that ssDNA stretching promotes strand separation.

4. Duderstadt, Chuang and Berger (2011) used a FRET-based assay to measure the binding of DnaA to dT-21, and suggested that DnaA stretches the ssDNA. The use of the term "stretch" is a little misleading. Compared with the flexibility of the free ssDNA, the increased distance measured by

FRET analysis simply reflects the greater rigidity of the complex of DnaA bound to the ssDNA.
Please consider using a different term.

We agree with the Reviewer on this point. However, here we have used the term stretching to describe the proposed mechanism of unwinding which takes into account both the FRET assay and the crystal structure where ssDNA stretching is directly observed.

5. Line 181 and 443. The 75 and 86 constructs are the only ones that work in both orientations.
Please qualify the statements in the respective lines.

The Reviewer is pointing out that the function of a DnaA-box at positions 75 and 86 was tested in both orientations, whereas several other locations were tested only in the rightward direction and one addition site (96) was tested only in the leftward direction (Fig 2). The key point we are trying to make is that the distal DnaA-box is able to function facing either leftward or rightward at multiple locations, not whether a DnaA-box at a specific site can always be reversed (this is an interesting question that can be addressed in future studies aimed at understanding the function of the proposed DNA loop in greater detail). We purposefully used the phrasing “it can function in opposite orientations” (line 189-190) and “can function in either orientation” (line 491) without mentioning any specific sites to draw only a general conclusion regarding the distal DnaA-box.

6. Line 204 and Figure S4. Isn't it possible that the increased abundance of DnaN is responsible? If so, please modify the statement in line 204.

This is a good point. To address this we created a strain with a xylose-inducible copy of *dnaA* and show that overexpression of DnaA alone also rescues activity of a minimal *incC* (DnaA-box#6/7), indicating that DnaN is not responsible (Fig EV1D-E). We have added this information to the text (line 213-215).

7. Line 235-236. Isn't the data simply consistent with the requirement for a few rounds of DNA replication preceding the commitment to spore formation?

Two chromosomes are required for a *B. subtilis* cell to complete endospore formation. Importantly, if DNA replication is incomplete or if the chromosomes are damaged then sporulation will be blocked. We have added this information to the text (line 237-239).

8. Figure 6F. In the lanes with DeltaDnaA-trios, the total amount of DnaA decreases with time. If DnaA is at the top of the gel and not visible in the panel, does this mean that the absence of DnaA-trios promotes concerted filament formation?

DnaA was not observed at the top of the membrane.

9. Line 129 and elsewhere. The IPGT-regulated promoter is a little leaky. See Bhavsar et al., (AEM (2001) 67:403). Please revise the statement.

The phrase “tightly regulated” has been removed.

10. Line 153. DnaA box 2/3 + 6 shows partial activity. Please revise the text.

The text now reads “Supporting this interpretation we were also able to construct a minimal *incC* with partial activity containing a distinct two subregion structure with DnaA-box#2/3 upstream and DnaA-box#6 downstream” (line 160-162).

11. Figure 2A. Did the authors test the constructs with the DnaA-box#1 and 4 in the other orientation?

No.

12. Figure 6F. Does the interpretation of the crosslinked species rely on molecular weight markers or are they deduced? If molecular weight markers were used, it may help to include their positions in Figure 6F.

The interpretation of the crosslinks is based on experiments where molecular weight markers were included (see image below).

13. Line 492-498 and elsewhere where serial dilutions are shown. Was the turbidity of each culture first adjusted so that they were equal before preparing the dilutions?

Overnight cultures were saturated and the total number of colony forming units of various strains was observed to be similar.

14. Figure 2B and 2D. What was the length of incubation?

Between 48 and 72 hours (see Methods).

15. Line 229. Isn't the decrease 100-fold (row 1 vs 2, Figure 3E)?

Changed (we had not accounted for the 10-fold decrease in total cells).

16. Line 276. F128A and E183A are not entirely wild type in activity.

The Reviewer is correct. However, we were focused on residues that were essential and both F128 and E183 retained significant activity.

17. Line 396. "a the"

Changed.

18. Line 719. Do the authors want to also label the DnaA-trios region in Figure 1a with "incC unwinding region?"

Good idea. Added.

19. Please consider removing Figure S1B.

We believe that this panel reinforces the fact that DnaA contains two different DNA binding activities: dsDNA binding for the DnaA-box and ssDNA binding for the DnaA-trios. We are happy to remove this at the Editor's discretion.

Referee #3:

1. There is an alternative interpretation consistent with these observations [analyzing the chimeric DnaA]. Given that the primary oligomerization domain of the chimera is the same as for wild-type *Bacillus subtilis* DnaA, it is also possible a mixed complex forms on boxes#6/7 composed of wild-type *Bacillus subtilis* DnaAs bound to boxes#6/7 and chimeric DnaAs co-assembling though domain III. In this case, the combined expression of wild-type and chimera could be sufficient to support origin unwinding without the need for box#CR (box#3) as demonstrated by the authors for higher levels of wild-type expression alone. To exclude this possibility the authors should do a control in which box#CR (box#3) is removed but both the chimera and the wild type are expressed and demonstrate this does not support origin unwinding.

A good point. We performed this experiment and found that the activity of the chimeric DnaA requires the upstream *T. maritima* DnaA-box. The data is provided in Figure 4C and described in the text (line 280-282).

Furthermore, in Fig 4D the authors use a strain with an endogenous *dnaA* KO to confirm chimeric mutant expression. Is the wild-type *dnaA* viable in such a genetic background when overexpressed either in presence of only box6+7 (according to Fig 3B) or with the additional box3 *T.m.* (Fig 4C)?

The $\Delta dnaA$ strain was only used to confirm expression of the chimeric proteins, not to evaluate their function. In light of this and the additional experiment showing that the chimeric DnaA requires the upstream *T. maritima* DnaA-box for its function, we believe that there is strong evidence to support the specificity of the chimeric DnaA proteins.

A decisive experiment demonstrating the chimera and wild-type DnaA bind only the desired boxes

hasn't been provided. The above experiments would clarify this issue. Alternatively, an EMSA could be used to confirm the specificity of the sites for the T.m. vs. wild-type DnaA variants.

Because we have only used the DnaA chimera *in vivo*, we believe that the experiment described above demonstrating the *T. maritima* DnaA-box is required for the activity of the chimeric DnaA is the most relevant to support the physiological DNA binding specificity.

2. Figure 5 and 6 show an *in vitro* unwinding assay on a short duplex substrate containing DnaA boxes#6/7 and the adjacent DnaA-trios. Addition of DnaA and a competing oligo for the trio region allows for unwinding. This authors show this activity depends on the DnaA binding sites and the trio. However, given the small size of the substrate it is hard to visualize how the adjacent DnaA molecules might engage with the trios. An alternative explanation is that unwinding is conducted *in trans* wherein one substrate bound DnaA oligomer engages with the trio of another substrate. To test this possibility the authors should conduct an experiment in which they mix a substrate with functional DnaA binding sites and corrupted trio with a substrate with corrupted DnaA binding sites and a functional trio. If unwinding still occurs in this case it would indicate complexes are acting *in trans*. Otherwise, unwinding occurs entirely within a single substrate as suggested in the manuscript. This is an insightful point that we have integrated into the manuscript (Figure 7A and line 375-390). To address this we performed the proposed mixing experiment and did not observe strand separation, suggesting that unwinding occurs within a single substrate. The data is provided in Figure 7B-C and described in the text (line 391-396).

Because the suggested experiment produced a negative result, we designed a further experiment to test these alternative models. Briefly, we captured biotin-labelled DNA scaffolds on streptavidin coated magnetic beads to decrease the effective concentration of DNA substrates in the reaction and thereby limit the potential for unwinding *in trans*. Here we observed that the unwinding rate was comparable whether the DNA was captured or in solution, again consistent with the model that unwinding occurs within a single substrate. The data is provided in Figure 7D-E and S6 and described in the text (line 397-406).

3. Fig 2B: Why were distance 86, 91 and 96 chosen for reversed box3 orientation? Does the helical phasing for the reversed orientation follow the same behavior as in Fig 2A?

Distances of 86 and 91 corresponded to positions of rightward facing DnaA-boxes, 96 is then 5 base pairs further upstream. We have not tested the behaviour of a leftward facing DnaA-box in the same detail, although we plan to perform such experiments to further understand the role of the proposed DNA loop.

4. Fig 2D: The DNA cartoon is fully aligned and looks the same scale as in Fig 2A & Fig 2B but the distances are much larger. Highlight the larger distance in a graphical way to clarify for the reader (e.g. by a scale break)

Good idea. Done.

5. The trio elements should be introduced more fully in the introduction. Without knowledge of previous work from the group their importance for origin function is not entirely clear.

We have expanded our introduction to include the following: “Domain III also contains the residues required for DnaA to interact specifically with a ssDNA binding site termed the “DnaA-trio” (Duderstadt *et al*, 2011; Ozaki *et al*, 2008; Richardson *et al*, 2016). The DnaA-trio is a repeating trinucleotide motif (consensus 3'-GAT-5') originally discovered as an essential element within the *Bacillus subtilis* origin and then identified in origins throughout the bacterial domain. It has been proposed that DnaA-trios stabilise a DnaA filament on a single DNA strand, with each DnaA-trio motif interacting with a single subunit of DnaA from the filament (Richardson *et al*, 2016).” (line 53-62)

6. The sentence between lines 180 and 184 should be split or shortened and focused for clarity.

We have modified the sentence as follows: “Taken together the following observations suggest that the distal DnaA-box might act through a DNA loop: (i) it can function at multiple locations; (ii) it can function in opposite orientations; (iii) it can function at great distances from the downstream subregion; (iv) its activity displays helical phasing (Bellomy *et al*, 1988; Hochschild *et al*, 1986)” (line 188-191).

7. The authors mention the oligomerization activity of domain I in the discussion as an alternative mode for two distant DnaA subcomplexes to co-assemble. This could be tested directly by removing domain I from the chimeric DnaA and observing if origin function is still supported. Alternatively, perhaps the authors could point out some examples from the literature supporting this role for domain I.

This is a good idea and is an experiment we are planning in the context of understanding the proposed DNA loop. As suggested we have now cited an example of *Helicobacter pylori* HobA, which interacts with domain I and promotes DnaA-dependent bridging of DNA molecules *in vitro* (Zawilak-Pawlik *et al*. 2007 Mol Micro) (line 484-486).

2nd Editorial Decision

5th June 2019

Thank you for submitting your revised manuscript to The EMBO Journal, and apologies for the delayed re-evaluation. We have now received re-reviews from the three original referees, with two of them copied below, and the (formatted) report of referee 2 attached separately to this email. By and large, the referees are satisfied with the revisions, but reviewers 1 and 2 retain a number of specific reservations regarding presentation and interpretation, which I feel should be addressed during a final round of minor revision. Please note that this shall include modification of the title, as requested by referee 2 and (in follow-up consultations) seconded by referee 1. Furthermore, referee 1 also agreed with the last three points of referee 2, while not feeling that further changes in response to points 2-4 would be essential (see additional comments below).

REFEREE REPORTS

Referee #1:

All major transactions on DNA require its strands to separate, an energetically unfavorable reaction that has been a challenge to understand ever since the birth of molecular biology. The paper reports an heroic accomplishment of reducing a complex origin opening system to a few elements in vitro, and set the stage for further in depth mechanistic studies. More importantly, this minimal system appears to have been conserved in all bacteria. The present information will surely encourage/guide DNA-opening in other systems. The general significance of the work cannot be any higher.

The authors have addressed all my concerns in the revised ms. and provided new data where appropriate.

I noted a few places where a bit more clarification might help. It is entirely up to the authors how they want to deal with these.

1. l.145: Any explanation for why Richardson et al found box#6 to be sufficient?
2. Fig. 6B: The text says "sequences scrambled" (l.357) but the figure is labeled with Δ DnaA-boxes and Δ DnaA-trios. Label the figure with scrambled rather than deletion? The same applies to Fig 6D & F.
3. L.368: Delete "in contrast".
4. Fig. 6H: The filament growth is cartooned as from the 5' end of lower strand towards 3' (see Fig. 5F). In that case why should one vs. three bp deletion matter? It would make more sense if the filament grows from the DnaA box end (as in Fig 7F). Please clarify the directionality of filament growth with respect to 3'GAT5'.
5. L.415: The statement that "it acts to deliver DnaA to DnaA-trios" may be a bit too direct since it is not clear where and how the loop closes. The following statement that "it acts to increase local---origin unwinding" is safer. Same comment on l.470.

Referee #2:

EMBOJ-2019-101649

Identification of a basal system for bacterial chromosome origin unwinding
PLEASE SEE ATTACHED FILE WITH FORMATTED REVIEW

ATTACHED FILE WITH FORMATTED REVIEW

EMBOJ-2019-101649

Identification of a basal system for bacterial chromosome origin unwinding

Please see the italicized text.

Referee #2:

1. Please consider revising the title. Although the contributions of individual DnaA boxes and DnaA-trio motifs in the replication origin of *Bacillus subtilis* have been analyzed to identify those that are necessary and sufficient for function in this organism, it is unclear whether the comparable sequences in the replication origins of other bacteria are also necessary and sufficient. The underlying problem with the title is the word "bacterial."

We appreciate this point. However, we believe the combination of functional data from *B. subtilis*, combined with bioinformatic evidence showing that these basal elements are conserved in diverse bacterial origins (Richardson *et al.* 2016 Nature), make a logical case for proposing that the basal system is widespread throughout the domain. This viewpoint is reflected by Reviewer#3: "The approach provides numerous novel insights into the organization of the *Bacillus subtilis* origin which, through comparative analysis, provide a framework for understanding the minimal conserved elements of all bacterial origins."

*In this reviewer's opinion, the role of the DnaA-trio motif supports a _model_ that the authors speculate is relevant to other bacteria. This reviewer remains concerned about the term "bacterial" in the title. An acceptable alternative may be to include the word "model" in the title, such as "A model of DnaA filament formation derived from studies of *Bacillus subtilis* DnaA." Of relevance, Reviewer 1 was concerned with "unwinding" in the title; this reviewer agrees with her/his comment.*

2. A reservation with the results is whether the minimal motifs of DnaA box 6 and 7 and the DnaA-trio region support synchronous initiation. Please consider adding a qualifying statement.

We have added a qualifying statement to the Discussion: "Based on the results in *B. subtilis* where only two of the seven DnaA-boxes within the unwinding region are essential, we speculate that the majority of DnaA-boxes within bacterial origins are either redundant or regulatory (i.e. – coordinating the timing or synchrony of origin firing within the cell cycle)" (line 448-451).

The problem with the revised text is the word, “essential.” The authors do not know the conditions under which other DNA sequences become “essential.” Please consider replacing “essential” with “important” or “critical.”

3. The publication by Duderstadt, Chuang and Berger (2011) suggests a model that ssDNA is stabilized by interaction with amino acids in the interior of the DnaA filament. One experiment in the paper is a strand displacement assay in which the binding of DnaA to a short (15mer) dsDNA fluorescently-labeled in one strand led to partial displacement of the fluorescently-labeled ssDNA. A 30mer was inert. Strand displacement activity has been also been ascribed to T4 SSB, E. coli SSB, and hRPA that bind to ssDNA cooperatively. Other than DnaA boxes 6 and 7 serving as a nucleation site, is the "unwinding assay" measuring strand displacement?

This is an interesting question. The proposed mechanism for how DnaA engages and stretches a single DNA strand suggests that it contacts the surface of the DNA backbone, rather than the surface of the bases, such that it would not be positioned to physically displace the complementary strand (Duderstadt *et al.* 2011 Nature). Mutagenesis *in vitro* and *in vivo* supports the crystallographic model (Duderstadt *et al.* 2011 Nature; Ozaki *et al.* 2008 JBC; this manuscript). Taken together we believe the data supports the model that ssDNA stretching promotes strand separation.

The authors' responses have not excluded the possibility that the assays measuring the dissociation of the Cy5-labeled oligonucleotide actually indicate strand displacement activity. The issue is especially important considering the use of the term “unwind” and its derivatives throughout the manuscript.

4. Duderstadt, Chuang and Berger (2011) used a FRET-based assay to measure the binding of DnaA to dT-21, and suggested that DnaA stretches the ssDNA. The use of the term "stretch" is a little misleading. Compared with the flexibility of the free ssDNA, the increased distance measured by FRET analysis simply reflects the greater rigidity of the complex of DnaA bound to the ssDNA. Please consider using a different term.

We agree with the Reviewer on this point. However, here we have used the term stretching to describe the proposed mechanism of unwinding which takes into account both the FRET assay and the crystal structure where ssDNA stretching is directly observed.

Please revise the text to define the term “DnaA-mediated stretching,”

which means that DnaA bound to dT-21 increases the rigidity of the ssDNA compared with the unbound ssDNA.

8. Figure 6F. In the lanes with DeltaDnaA-trios, the total amount of DnaA decreases with time. If DnaA is at the top of the gel and not visible in the panel, does this mean that the absence of DnaA-trios promotes concerted filament formation?

DnaA was not observed at the top of the membrane.

Where is DnaA then? Is it at the top of the gel, and then lost when the gel is placed onto the membrane? If so, the conclusion that “mutating the DnaA-trios abrogated DnaA^{cc} filament formation” in line 362-363 is not correct.

12. Figure 6F. Does the interpretation of the crosslinked species rely on molecular weight markers or are they deduced? If molecular weight markers were used, it may help to include their positions in Figure 6F.

The interpretation of the crosslinks is based on experiments where molecular weight markers were included (see image below).

If the authors have not already described how they deduced the composition of the crosslinked species in Figure 5E, and 6F, please add a statement to the manuscript.

14. Figure 2B and 2D. What was the length of incubation?

Between 48 and 72 hours (see Methods).

It would have been appropriate to describe the specific incubation periods in the respective legends.

Referee #3:

The revised manuscript by Richardson et al. has addressed my main concerns. In particular, the authors excluded the possibility of a mixed WT/Chimeric complex forming on boxes#6/7 alone by omitting box#CR. This lends further support for their conclusion that a loop forms between the binding regions in their minimal origin.

The additional experiments provided in Figure 7 very nicely exclude the possibility of trans unwinding by different DnaA complexes. The biotin pulldown had not occurred to me, but is very convincing.

It would have been nice to see additional experiments involving domain I given that domain I is known to oligomerize and this activity may provide additional pathways for assembly and loop formation between origin elements. But I understand this may be a more substantial study beyond the scope of the work presented.

Finally, the authors have corrected several minor display issues and confusing sections in the text. These revisions have significantly improved what was already a very insightful paper about the minimal requirements for origin unwinding. This work brings us one step closer to understanding what ultimately defines a minimal functional origin.

REFEREE 1 COMMENTS ON REFEREE 2's CONCERNS:

1. I agree with the Referee. The title should have an "a" before bacterial (e.g., "Identification of a basal system for unwinding of a bacterial chromosome origin". The applicability of the basal system hasn't been shown even for a second bacterium yet. The claim of generality from bioinformatics has been already published in an earlier paper (Nature 2016). sal system. Now is the time for experimental demonstration.
2. The authors' reply is fine.
3. and 4. The authors' reply is fine. Stretching and strand displacement cannot happen without unwinding. The reviewer is suggesting using a vague term "rigidity" rather than 'stretching' which has been observed in co-crystals of ssDNA and DnaA, and called that way by no other than James Berger. There is an obligation to stick to terms that has been already introduced, unless proven wrong.
8. The referee has a point here. The assertion that total amount of DnaA decreases with time may not be correct. The authors should quantify the monomer band (in lesser exposed) gels. Also, they should explain why the filaments are not seen when the reaction mixture is incubated for longer times (Fig 5F and 6F).
12. I agree with the Referee.
14. I agree with the Referee.

2nd Revision - authors' response

6th June 2019

RESPONSE TO REVIEWERS

Referee #1:

1. L. 145: Any explanation for why Richardson et al found box#6 to be sufficient?

Apologies, we are not sure what the Reviewer is asking. DnaA-box#6 was previously shown to be necessary (see Richardson *et al.* 2016 Nature). Here we show it is not sufficient (Figure 1C).

2. Fig. 6B: The text says "sequences scrambled" (l.357) but the figure is labeled with Δ DnaA-boxes and Δ DnaA-trios. Label the figure with scrambled rather than deletion? The same applies to Fig 6D & F.

Changed.

3. L.368: Delete "in contrast".

Changed.

4. Fig. 6H: The filament growth is cartooned as from the 5' end of lower strand towards 3' (see Fig. 5F). In that case why should one vs. three bp deletion matter? It would make more sense if the filament grows from the DnaA box end (as in Fig 7F). Please clarify the directionality of filament growth with respect to 3'GAT5'.

We have modified all cartoons to indicate filament growth on the lower strand in the 3'→5' direction.

5. L.415: The statement that "it acts to deliver DnaA to DnaA-trios" may be a bit too direct since it is not clear where and how the loop closes. The following statement that "it acts to increase local--origin unwinding" is safer. Same comment on l.470.

L415 changed to "Characterization of the distal subregion suggests that it acts to increase the local concentration of DnaA at the site of origin unwinding" and L470 changed to "In contrast to LacI, the experiments here using the DnaA chimera suggest that the replication initiator bound at the distal DnaA-box requires ssDNA binding activity."

Referee #2:

1. In this reviewer's opinion, the role of the DnaA-trio motif supports a_model_that the authors speculate is relevant to other bacteria. This reviewer remains concerned about the term "bacterial" in the title. An acceptable alternative may be to include the word "model" in the title, such as "A model of DnaA filament formation derived from studies of Bacillus subtilis DnaA." Of relevance, Reviewer 1 was concerned with "unwinding" in the title; this reviewer agrees with her/his comment.

Referee #1 Comment: I agree with the Referee. The title should have an "a" before bacterial (e.g., "Identification of a basal system for unwinding of a bacterial chromosome origin". The applicability of the basal system hasn't been shown even for a second bacterium yet. The claim of generality from bioinformatics has been already published in an earlier paper (Nature 2016). Now is the time for experimental demonstration.

We appreciate this point. We have modified the title to be "Identification of a basal system for unwinding a bacterial chromosome origin"

2. The problem with the revised text is the word, "essential." The authors do not know the

conditions under which other DNA sequences become “essential.” Please consider replacing “essential” with “important” or “critical.”

Referee #1 Comment: The authors' reply is fine.

We have not modified the statement.

3. The authors' responses have not excluded the possibility that the assays measuring the dissociation of the Cy5-labeled oligonucleotide actually indicate strand displacement activity. The issue is especially important considering the use of the term “unwind” and its derivatives throughout the manuscript.

4. Please revise the text to define the term “DnaA-mediated stretching,” which means that DnaA bound to dT-21 increases the rigidity of the ssDNA compared with the unbound ssDNA.

Referee #1 Comment: The authors' reply is fine. Stretching and strand displacement cannot happen without unwinding. The reviewer is suggesting using a vague term "rigidity" rather than 'stretching' which has been observed in co-crystals of ssDNA and DnaA, and called that way by no other than James Berger. There is an obligation to stick to terms that has been already introduced, unless proven wrong.

We have not modified the text.

8. Figure 6F, scrambled DnaA-trios. Where is DnaA then? Is it at the top of the gel, and then lost when the gel is placed onto the membrane? If so, the conclusion that “mutating the DnaA-trios abrogated DnaAcc filament formation” in line 362-363 is not correct.

Referee #1 Comment: The referee has a point here. The assertion that total amount of DnaA decreases with time may not be correct. The authors should quantify the monomer band (in lesser exposed) gels. Also, they should explain why the filaments are not seen when the reaction mixture is incubated for longer times (Fig 5F and 6F).

In this case the process of transferring the protein from the gel to the membrane has caused the sample to spread, creating the appearance that the amount of DnaA is being lost. Quantification of the monomer in the two samples shows that there is a similar trend in both reactions over time (Wild-type % of monomer relative to 0 min: 100, 88, 69, 71 vs Scrambled DnaA-trios % of monomer relative to 0 min: 100, 80, 75, 71).

It is unclear why DnaA filaments are not seen when the reaction mixture is incubated for longer times. We hypothesize that the cysteine residues may become oxidized and therefore do not react with the crosslinker, but we have not investigated this further.

8. If the authors have not already described how they deduced the composition of the crosslinked species in Figure 5E, and 6F, please add a statement to the manuscript.

Referee #1 Comment: I agree with the Referee.

We have added the following statement to the Methods: “The interpretation of the crosslinked species is based on their migration relative to molecular weight markers” (line 635-636).

14. It would have been appropriate to describe the specific incubation periods in the respective legends.

Referee #1 Comment: I agree with the Referee.

We have now indicated all incubation periods within the Figure panels.

Referee #3:

1. It would have been nice to see additional experiments involving domain I given that domain I is known to oligomerize and this activity may provide additional pathways for assembly and loop formation between origin elements. But I understand this may be a more substantial study beyond the scope of the work presented.

An insightful point – we are currently performing these experiments and many others to better understand the interaction between subsets of DnaA-boxes. Indeed these will constitute the basis of an independent study.

Accepted

7th June 2019

Thank you for submitting your final revised manuscript for our consideration. I am pleased to inform you that we have now accepted it for publication in The EMBO Journal.

Corresponding Author Name: HEATH MURRAY

Journal Submitted to: EMBO JOURNAL

Manuscript Number: EMBOJ-2019-101649